



# Multi-model analysis of the radiative impacts of the 2022 Hunga eruption indicates a significant cooling contribution from the volcanic plume

Ilaria Quaglia[1], Daniele Visioni[1,2], Ewa M. Bednarz[3,4], Yunqian Zhu[3,4], Georgiy Stenchikov[5], Valentina Aquila[6], Cheng-Cheng Liu[7], Graham W. Mann[8], Yifeng Peng[9], Takashi Sekiya[10], Simone Tilmes[1], Xinyue Wang[11], Shingo Watanabe[12], Pengfei Yu[13], Jun Zhang[1], and Wandi Yu[14]

[1]NSF National Center for Atmospheric Research, Boulder, CO, USA
[2]Department of Earth and Atmospheric Sciences, Cornell University, USA
[3]Cooperative Institute for Research in Environmental Sciences (CIRES), University of Colorado, Boulder, USA
[4]NOAA Chemical Sciences Laboratory, Boulder, USA
[5]Physical Science and Engineering Division, King Abdullah University of Science and Technology, Jeddah, Saudi Arabia
[6]American University, Department of Environmental Science, Washington, DC, USA
[7]Laboratory for Atmospheric and Space Physics, University of Colorado Boulder, Boulder, CO, USA
[8]School of Earth and Environment, University of Leeds, Leeds, UK
[9]Lanzhou University, Lanzhou, China
[10]Japan Agency for Marine-Earth Science and Technology (JAMSTEC), Yokohama, Japan
[11]Department of Atmospheric and Oceanic Sciences, University of Colorado Boulder, Boulder, USA
[12]Advanced Institute for Marine-Ecosystem Change, Tohoku University, Sendai, Japan
[13]Jinan University, Guangzhou, China
[14]Lawrence Livermore National Laboratory, USA

**Correspondence:** Ilaria Quaglia (iquaglia@ucar.edu)

**Abstract.** On January 15, 2022, the Hunga volcano eruption released unprecedented amounts of water vapor into the atmosphere alongside a modest amount of $SO_2$. In this work we analyse results from multiple Earth system models as part of the Hunga Tonga-Hunga Ha'apai Volcano Impact Model Observation Comparison Project.

Our results show a good model agreement over the climatic outcomes of the eruption, overall indicating a significant negative

radiative forcing from the Hunga eruption. The multi-model mean of global instantaneous radiative forcing averaged over 2022-2023 is estimated at -0.19 $\pm$ 0.04 W/m$^2$ at the top-of-atmosphere (TOA), and -0.16 $\pm$ 0.03 W/m$^2$ at the surface. Simulations with free-running meteorology and climatological sea surface temperatures and sea ice yield a global mean TOA forcing of -0.14 $\pm$ 0.10 W/m$^2$ across two models for the first 2 years, decreasing to -0.09 $\pm$ 0.10 W/m$^2$ on average between 2022 and 2027. However, these global values may be underestimated by about 50%, considering that recent $SO_2$ injection retrievals

suggest nearly twice the amount than the 0.5 Tg-$SO_2$ used in the protocol. We also find that the contribution from added stratospheric water vapor is minimal and that the injected $SO_2$ and the resulting formation of stratospheric sulfate dominate the radiative forcing. However, water vapor played a key role in the initial aerosol growth, leading to a stronger negative radiative forcing during the first six months after the eruption compared to simulations without water vapor co-injection.



## 1   Introduction

Explosive volcanic eruptions that release large amounts of $SO_2$ into the stratosphere have long been identified as a significant contributor to the global energy budget (Schmidt et al., 2018). The release of $SO_2$ into the stratosphere results in the formation of sub-micron, supercooled sulfate aerosols which efficiently reflect incoming solar radiation, resulting in a negative forcing at the top of the atmosphere and thus a corresponding surface cooling (Kremser et al., 2016). While often studies of volcanic aerosols' impact on climate focus on very large eruptions like Mt. Pinatubo, that erupted in 1991 (Quaglia et al., 2023) releas-

ing anywhere between 10 and 20 Tg-$SO_2$ in a few days (Baran and Foot, 1994; Fisher et al., 2019), the most recent decades have seldom seen eruptions with injections larger than a few Tg of $SO_2$ (Carn et al., 2017; Schmidt et al., 2018; Brodowsky et al., 2021). While smaller than Pinatubo, even one "moderate" eruption can double the quiescent stratospheric sulfate burden (Brodowsky et al., 2024), and their analyses can provide very important insights into aerosol microphysics, plume evolution and associated top-of-atmosphere radiative forcing (Brodowsky et al., 2021; Li et al., 2023).

The co-emission of other byproducts of volcanism, together with sulfate, is not infrequent: some ash and water vapor usually reaches the stratosphere (Zhu et al., 2020), and in some occasions chlorine and bromine in significant quantities are also co-injected, but whether they reach the stratosphere are debatable (Staunton-Sykes et al., 2021). Hunga Tonga-Hunga Ha'apai (Hunga) erupted on January 15 2022 in the South Pacific, releasing unprecedented amounts of materials into the atmosphere

(Carr et al., 2022). Current estimates of injected material are $146 \pm 5$ Tg of water vapor based on Aura Microwave Limb Sounder (MLS) retrievals (Millán et al., 2022), with a broader range for injected $SO_2$ between $0.41 \pm 0.02$ Tg of $SO_2$ for MLS and $1.6 \pm 0.5$ Tg of $SO_2$ for the Infrared Atmospheric Sounding Interferometer (IASI) (Sellitto et al., 2024). While the amount of injected sulfur was within the range of many minor past explosive volcanic eruptions, the amount of water vapor co-injected was unprecedented in the historical record, representing roughly $\sim$10% of all background stratospheric water vapor. The height

of the injected $SO_2$ was also, by many standards, unprecedented, reaching a height of 55 km (Carr et al., 2022). Early estimates assumed the water vapor forcing would result in a small net warming (Jenkins et al., 2023). However, other estimates found that the sulfate aerosols produced by the co-injections of $SO_2$, which normally oxidize to form sulfuric-acid aerosols, would result in a significant negative forcing after the plume spread uniformly (Sellitto et al., 2022; Zhu et al., 2022; Schoeberl et al., 2024), larger than the water vapor forcing. Later modeling analyses by Stenchikov et al. (2025), using WRF-chem, also found

a negative forcing by considering similar factors. Notably, the increased size of aerosols formed under conditions of significant stratospheric hydration (Zhu et al., 2022) has been suggested as an important factor contributing to the significant radiative impact, even in the case of a modest $SO_2$ injection.

As part of the Hunga Tonga-Hunga Ha'apai (HTHH) Impacts activity that was established in the World Climate Research

Programme (WCRP) Atmosphere Processes And their Role in Climate (APARC), a multi-model-observation intercomparison named the Hunga Tonga-Hunga Ha'apai Volcano Impact Model Observation Comparison (HTHH-MOC) Project was proposed in Zhu et al. (2024) with the aim to better ascertain the radiative and climatic impacts of Hunga in a multi-model context and





to try and separate the volcanic impacts from other natural or anthropic perturbations in the same years (Forster et al., 2023),
especially in light of the "unprecedented" surface temperature warming in 2023 (Cattiaux et al., 2024; Quaglia and Visioni,
2024). Other parallel works within this same project discuss in depth changes to stratospheric water vapor, aerosols, temper-
ature, and ozone (Zhuo et al., 2025) and potential impacts on climate (Bednarz et al., 2025). In this paper, we focus on the
radiative forcing arising from these changes by some of the proposed HTHH-MOC experiments in a multi-model context,
highlighting sources of agreement across models as well as differences in the estimated forcing magnitude.

## 2  Methods

### 2.1  Models and simulations

We use a suite of experiments described in depth in Zhu et al. (2024), which includes also a detailed descriptions of the
models under analyses. The first set of experiments is a 2-year experiment over 2022-2023 with nudged temperature and
meteorology as well as observed sea surface temperatures (SSTs) and sea ice (Exp2a in Zhu et al. (2024), here denoted
"Nudged"; note this is distinct from Exp2b, not included in this study, which does not require prognostic aerosols). The
second set of experiments is a 10-year long experiment over 2022-2031 with free running meteorology, using either imposed
climatological SSTs and sea ice (Exp1_fixedSST in Zhu et al. (2024), here "Fixed-SST") or with the atmosphere model coupled
with the ocean (Exp1_coupled, here "Coupled"). Each set of experiments consists of simulations with 0.5 Tg of $SO_2$ and 150
Tg of $H_2O$ injections ("SO2andH2O") as well as control simulations without any injections ("NoVolc"). Some models also
performed additional single forcing experiments with injection of only $SO_2$ ("SO2only") or only $H_2O$ ("H2Oonly"). The
location, altitude, and amount injected are different between models in order to better match the observed plume in the first
couple of days, but they remain consistent across experiments between models. Detailed injection settings for each model are
provided in Table 1.

Nudged was conducted by 5 models: the Whole Atmosphere Community Climate Model version 6 (WACCM6; Gettelman
et al., 2019; Davis et al., 2023) coupled with the four-mode modal aerosol module (MAM4, Liu et al., 2012, 2016; Mills
et al., 2016) and the Community Aerosol and Radiation Model for Atmospheres (CARMA; Tilmes et al., 2023), WACCM6-
MAM and WACCM6-CARMA, respectively; the Model for Interdisciplinary Research On Climate - CHemical Atmospheric
general circulation model for Study of atmospheric Environment and Radiative forcing version 6 (MIROC-CHASER; Sekiya
et al., 2016); the atmospheric component of CESM1, the Community Atmosphere Model version 5 (CAM5; Lamarque et al.,
2012) using the sectional aerosol microphysics model CARMA (CAM5-CARMA; Yu et al., 2015), and the UK Earth System
Model version 1.1 (UKESM; Mulcahy et al., 2023). Fixed-SST was carried out with two models, WACCM6-MAM and
MIROC-CHASER, while only WACCM6-MAM participated in Coupled. WACCM6-MAM ran 30-member ensembles for
both Fixed-SST and Coupled, whereas MIROC-CHASER ran 10-member ensembles for Fixed-SST. Further information about
the participating models can be found in the references above, as well as in Zhu et al. (2024).




**Table 1.** Injection parameters. Adapted from Table 7 in Zhu et al. (2024)

| Model | $H_2O$ injected | $H_2O$ altitude | $SO_2$ injected | $SO_2$ altitude | Injection location |
|---|---|---|---|---|---|
| WACCM6-MAM | 150 Tg | 25-35 km | 0.5 Tg | 20-28 km | 22-14°S, 182-186°E |
| WACCM6-CARMA | 150 Tg | 25-35 km | 0.5 Tg | 26.5-36 km | 22-6°S,182.5-202.5°E |
| MIROC-CHASER | 150 Tg | 25-30 km | 0.5 Tg | 25-30 km | 22-14°S, 182-186°E |
| CAM5-CARMA | 150 Tg | 25-35 km | 0.5 Tg | 20-28 km | 22-14°S, 182-186°E |
| UKESM | 150 Tg | 25-30 km | 0.5 Tg | 25-30 km | 22-14°S, 182-186°E |

## 2.2 Radiative forcing estimations

There are important differences between estimates of radiative forcing of volcanic eruptions derived from nudged simulations - referred to as instantaneous radiative forcing (IRF) - , that from free-running atmosphere-only simulations - conventionally termed effective radiative forcing (ERF), following the definition by Forster et al. (2016) - and that form fully coupled simulations with interactive ocean - referred to generally as radiative forcing (RF).

IRF represents the combined effect of direct interactions between the forcing agent and radiation, as well as interactions between the forcing agent and clouds (Smith et al., 2018). ERF, defined by Myhre et al. (2013) as "change in the net TOA downward radiative flux after allowing for atmospheric temperatures, water vapour and clouds to adjust, but with surface temperature or a portion of surface conditions unchanged", is the sum of IRF and rapid adjustments. These rapid adjustments which occur over weeks to months, before global-mean surface temperatures can respond, are due to changes in tropospheric and stratospheric temperature, water vapor, surface albedo, and clouds, and are distinct from slower feedbacks that are driven by surface temperature changes (Smith et al., 2018; Sherwood et al., 2015). In coupled model simulations, RF includes the instantaneous radiative forcing (IRF), rapid adjustments to that forcing, and slower climate feedbacks resulting from the coupled ocean–atmosphere response (Chung and Soden, 2015).

IRF is calculated in climate models using a double radiation call that excludes aerosols from online radiative calculations ("Clean-Sky") following the method proposed in Stenchikov et al. (1998) in order to separate the contribution of aerosols from that of other components. Since nudging reduces variability in meteorological fields, it typically limits any stratospheric temperature adjustments, as radiative forcing is calculated as the difference between the perturbed and unperturbed case. As a result, the IRF from aerosols and water vapor, either combined or individually, can be approximated using the corresponding Nudged experiments. In the case of WACCM6-MAM simulations, both methodologies have been applied. Notably, in contrast to previous studies, all forcing estimates presented here treat stratospheric water vapor as a forcing rather than a feedback, due





to its direct injection.

In summary, the following sections explain how the different radiative forcing estimates are calculated. The term $F_{control}$
refers to background simulations without any injection, while $F_{volc}$ is used to indicate simulations that include volcanic forcing. Radiative forcing is calculated under both Clear-Sky (CS, without clouds) and All-Sky (AS, including the effects of clouds) conditions. Unless otherwise specified, all values are assumed to be calculated under Clear-Sky conditions.

$RF = F_{volc}^{clear} - F_{control}^{clear}$: Clear-Sky forcing from Coupled, including both injections of water vapor and SO$_2$.

$ERF = F_{volc}^{clear} - F_{control}^{clear}$: Clear-Sky forcing from Fixed-SST, including both injections of water vapor and SO$_2$.

$IRF = F_{volc}^{clear} - F_{control}^{clear}$: Clear-Sky forcing from Nudged-SO2andH2O, including both injections of water vapor and SO$_2$.

$IRF_{aerosol} = F_{volc}^{clear} - F_{control}^{clear}$: Clear-Sky forcing from Nudged-SO2only, including only injections of SO$_2$.

$IRF_{gas} = F_{volc}^{clear} - F_{control}^{clear}$: Clear-Sky forcing, including only injections of H$_2$O in Nudged-H2Oonly.

$F^{clean}$ refers to Clean-Sky calculations (excluding aerosols) and is available only in WACCM6-MAM, where it is derived
using a double radiation call. In this configuration, a second estimate of the instantaneous radiative forcing from aerosols and gases separately, as calculated below:

$$IRF_{aerosol} = (F_{volc}^{clear} - F_{volc}^{clean,clear}) - (F_{control}^{clear} - F_{control}^{clean,clear})$$

$$IRF_{gas} = F_{volc}^{clean,clear} - F_{control}^{clean,clear}$$


We calculate RF at three key atmospheric levels - top of the atmosphere (TOA), tropopause (TROP), and surface (SURF) - as the sign and magnitude of RF can differ by altitude and carry distinct physical implications. Radiative forcing at TOA reflects the overall perturbation to the Earth's energy budget and is commonly used to estimate the potential influence on global average temperature. At the tropopause, RF captures the net energy change affecting the coupled troposphere–surface
system, and is considered less affected by upper stratospheric processes and better represents tropospheric heating. In contrast, RF at the surface does not directly correspond to surface temperature responses but is more relevant for understanding impacts on the hydrological cycle, particularly changes in precipitation patterns (Ramaswamy et al., 2018).





## 3    Results and discussion

We present first the results based on the nudged simulations (Section 3.1), as this experiment was done by more models (five)
and the use of observed meteorological conditions also allows us to better distinguish the overall radiative impact by remov-
ing natural variability. Since the same nudging (of temperature and horizontal wind) was applied in both the control and the
volcanic injection experiments, taking the difference between the two simulations, as done in Fig. 1, isolates the direct radia-
tive forcing from the volcanic material, as any temperature response cancels out. Following that, we provide analyses of the
two models which provided the 10-year long free-running simulations with prescribed climatological SSTs and sea ice: these
analyses allow us to understand the long-term behavior of the forcing as well as include the combined chemical and dynamical
impacts and temperature adjustments. Finally, we complete those with an analyses of fully-coupled simulations in WACCM6-
MAM. In this case, we compare the results from the three different types of simulations within the same model, coupled with
analyses of the double-radiation call described in the methods above, to discuss and quantify the different contribution to the
forcings analyses elsewhere.

### 3.1    Multi-model comparison of instantaneous radiative forcing in the nudged simulations

Figure 1 shows a timeseries of zonal mean TOA IRF under clear-sky conditions from the nudged simulations. IRF accounts for
all optically active components, including stratospheric aerosols, water vapor, ozone, and polar stratospheric clouds (PSCs).
All five models show qualitative agreement in the spatial distribution of the forcing, with the response primarily located in the
Southern Hemisphere (SH). Therefore, most of the following analyses will focus on the SH only. The spatial pattern of negative
forcing, initially peaking in the tropics during the first few months and then moving to the mid-to-high southern latitudes by
2023, is consistent across models. Likewise, all models simulate a substantial negative forcing persisting through the end of
2023, and show similar results in attributing forcing primarily to aerosols (second column in Fig. 1), with mostly negligible
responses seen under only the $H_2O$ injection (third column in Fig. 1).

Global and hemispheric means are shown in Fig. 2 as multi-model averages, with one standard deviation indicating the inter-
model spread. At the TOA, models generally show a weak and negative water vapor forcing (as diagnosed from the H2Oonly
simulations). At the tropopause (second column in Fig. 2), the radiative forcing from water vapor is indeed positive but very
small. Regarding the aerosol forcing, which thus constitutes essentially the entirety of the Hunga forcing perturbation, models
generally show good agreement over the time evolution of the forcing but with some substantial differences in its magnitude.
The multi-model mean TOA IRF is -0.35 $\pm$ 0.08 W/m$^2$ averaged over 60°S to 0° for the period 2022-2023, and -0.26 $\pm$ 0.07
W/m$^2$ over the Southern Hemisphere high latitudes, averaged from September 2022 to December 2023.





Figure 3 provides further insight into the TOA IRF contributions from aerosols ($IRF_{aerosol}$) and water vapor ($IRF_{gas}$),based
on analyses from WACCM6-MAM using both methods described in Section 2.2. The figure also includes a comparison of
clear-sky and all-sky IRF. Multiple interesting features emerge from an analyses of Fig. 3.

- While the IRF from the SO2andH2O and SO2only simulations show a consistent behavior in the late part of 2022 and
  after, significant differences (in the order of 20%) can be found in the first six months or so (black and orange lines in
  panels 3a-c). This is in agreement with previous works which suggested that the large quantities of water vapor favored
  particle growth in the initial months (Zhu et al., 2022), resulting in a larger forcing. The fact that the forcing becomes
  similar later on, and shows a slight reduction in the last few months of 2023, may suggest a balance between larger
  particles and shorter lifetime for the two cases.

- A comparison between the aerosol IRF ($IRF_{aerosol}$; labeled as "Aerosol" in the figure) and the total IRF indicates that
  $IRF_{aerosol}$ accounts for approximately 80% of the overall IRF response, with the remaining contribution attributed to
  changes in other stratospheric gases (see panel 3b vs 3a; time-averaged values are summarized in panels 3d–e).

- H2Oonly simulations do not show any significant forcing in clear-sky, and a more negative, but also highly variable,
  forcing in all-sky (blue lines in panels 3a-c). This further confirms that the water vapor forcing alone does not result in
  either a cooling or a warming. Interestingly, a comparison of the $IRF_{gas}$ in the SO2andH2O and H2Oonly experiments
  (black and blue lines in panel 3c) suggests that the all-sky contribution is not coming from water vapor itself, but from
  changes to other radiatively-active gases. This will be further discussed in Section 3.3.

- The estimation of $IRF_{aerosol}$ and $IRF_{gas}$ differs depending on whether they are derived from the SO2-only and H2O-
  only experiments, respectively, or from the double radiation call method applied to SO2andH2O experiment (panels
  3d-e). Under both clear-sky (CS) and all-sky (AS) conditions, during the first 6 months after the eruption, $IRF_{aerosol}$
  and $IRF_{gas}$ obtained from the SO2only and H2Oonly experiments, respectively, are smaller than those calculated using
  the double radiation call in the SO2andH2O experiment (green and red boxes, respectively). This discrepancy occurs
  because, in the SO2andH2O experiment, water vapor influences aerosol growth, thereby affecting $IRF_{aerosol}$, while
  any simultaneous changes in other gases (e.g. ozone) will also contribute to $IRF_{gas}$.

## 3.2 Comparison of the effective radiative forcing in the free running simulations

Fig. 4 shows the longer-term evolutions of the moving average forcing (calculated as $\frac{1}{t}\int_0^t RF(t)dt$) in the free-running simu-
lations with fixed climatological SSTs and sea ice (Fixed-SST). For this experiment, only two models (WACCM6-MAM and
MIROC-CHASER) performed the simulations, and we show the moving average forcing instead of the instantaneous forcing to
highlight the cumulative forcing produced by the eruption over time (which is a better measure of its overall climatic impacts).
The TOA ERF estimates show a very good agreement between the two models, both in terms of the overall magnitude and the
rate of dissipation of the forcing, which is partly true also for the ERF estimates at the surface (see Tables A1 and A3). Larger
differences between the models are present for the ERF estimates at the tropopause, whereby WACCM6-MAM shows a more







**Figure 1.** Time series of zonal mean radiative forcing at the top-of-atmosphere (TOA) under clear-sky conditions from five models: WACCM6-MAM (a–c), WACCM6-CARMA (d–f), MIROC-CHASER (g–i), CAM5-CARMA (j-l), and UKESM (m). Each column corresponds to a different perturbation scenario from the nudged experiment: the first column shows SO2andH2O, the second column SO2only, and the third column H2Oonly.



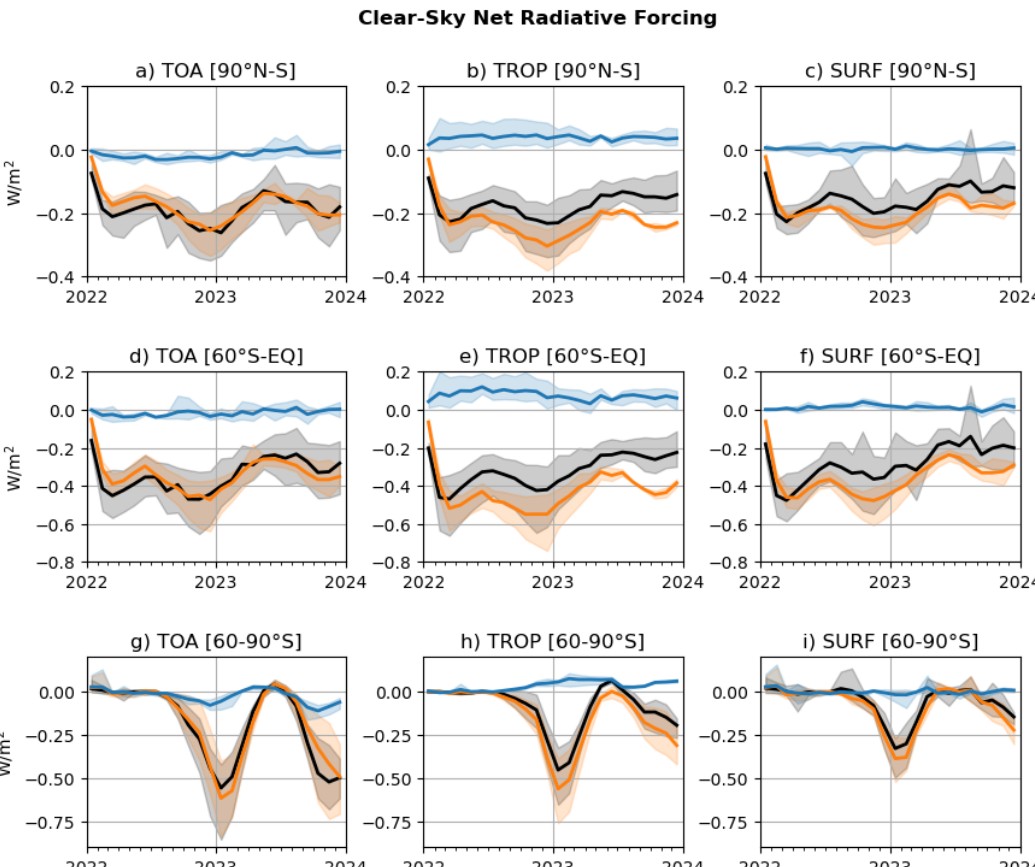

**Figure 2.** Time series of net radiative forcing under clear-sky conditions at three atmospheric levels: TOA (a, d, g), TROP (b, e, h), and SURF (c, f, i). Solid lines indicate the multi-model means for the nudged experiments as indicated in the legend, while the shaded areas represent the range (minimum to maximum) across models. Each row corresponds to a different latitudinal averaging region: the first row covers 60°S to 90°N, the second 60°S to the equator, and the third 60–90°S.

negative average forcing (by 0.1 W/m$^2$) in the first year (Table A2) and a different overall trend throughout the simulations.

Although the ERF response is broadly consistent between the two models, notable differences emerge in their simulated stratospheric aerosol optical depth (sAOD). In particular, we observed significant discrepancies in the timing of aerosol for-
mation and the onset of its decline. In MIROC-CHASER, stratospheric AOD starts to decline two months after the eruption, right after reaching its peak. In contrast, WACCM6-MAM shows a peak in stratospheric AOD three months post-eruption, which remains at peak levels for six months before beginning to decrease. However, the models generally agree on the plume's southward transport, with the plume moving towards 60°S after 4 to 5 months and reaching the pole by December 2023. The



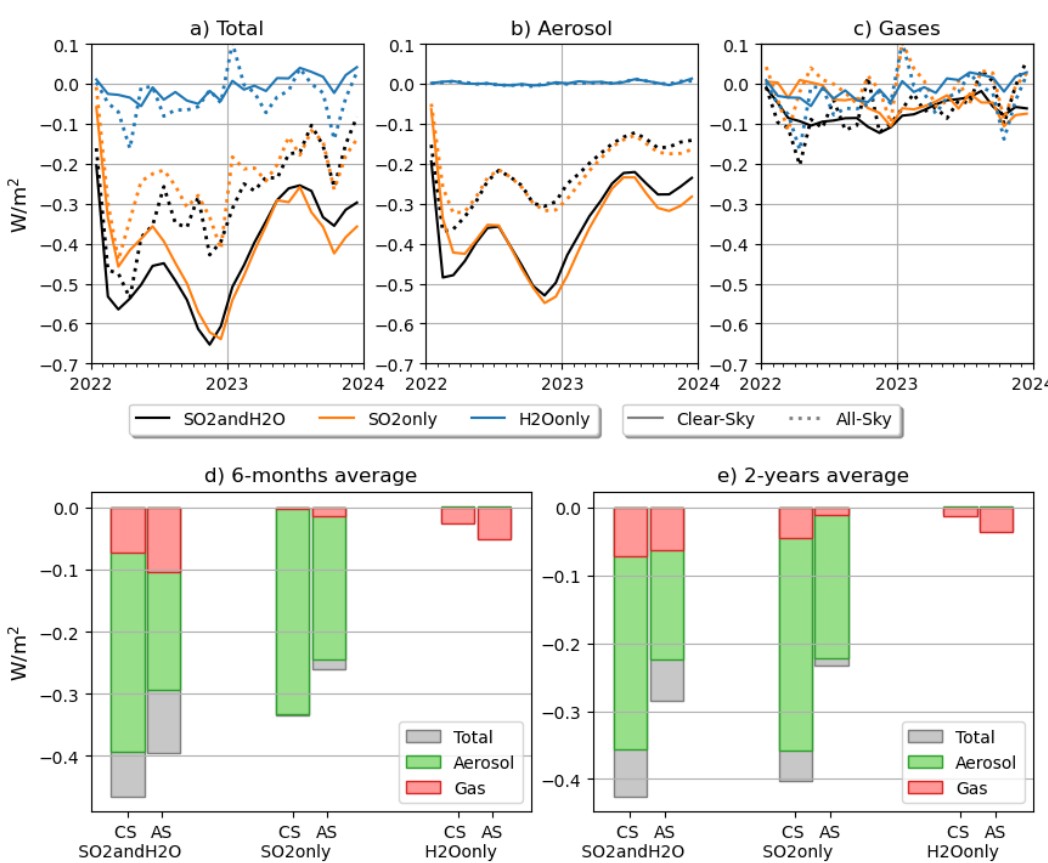

**Figure 3.** Time series of net radiative forcing at TOA, averaged between 60°S and the equator, in WACCM6-MAM. Panel (a) shows the total IRF (aerosols+gases), (b) shows the aerosol-only contribution, and panel (c) shows the gas-only contribution, all derived using the double radiation call method (see Section 2.2). Different colors indicate various nudged experiments, with solid lines representing clear-sky conditions and dotted lines indicating all-sky conditions. Panels (d) and (f) show the 6-month and 2-year averages, respectively, of the radiative forcing from the three nudged experiments, under clear-sky (CS) and all-sky (AS) conditions. The contributions from aerosols and gases alone are highlighted relative to the total forcing.

persistence of the aerosol plume at low-to-mid latitudes in the SH is different between the two models, with WACCM6-MAM
showing some persistence all the way to the end of 2023, whereas MIROC-CHASER show a full dissipation of the cloud by
late 2023. Contrary to sAOD, the water vapor upward diffusion shows more persistence in MIROC-CHASER than WACCM6-
MAM, with the latter showing no residual water vapor anomaly below 1 hPa by late 2026, whereas MIROC-CHASER shows at
least 0.5 ppm more in the upper stratosphere remaining all the way to 2028. MIROC-CHASER also shows a stronger positive
tropical ozone anomaly between 10 and 20 hPa compared to WACCM6-MAM, and a negative ozone anomaly higher up be-
tween 1 and 2 hPa, with the negative anomaly more persistent over time in both models, explained by a potential enhancement



**Table 2.** Effective radiative forcing (in W/m$^2$) at the Top of Atmosphere under clear-sky conditions. Forcing is averaged between 60°S and the equator.

| Model | 6 Months | 1 Year | 2 Years | 5 Years | 10 Years |
|---|---|---|---|---|---|
| WACCM6-MAM | $-0.31 \pm 0.14$ | $-0.36 \pm 0.14$ | $-0.27 \pm 0.14$ | $-0.12 \pm 0.14$ | $-0.07 \pm 0.06$ |
| MIROC-CHASER | $-0.30 \pm 0.17$ | $-0.35 \pm 0.11$ | $-0.24 \pm 0.12$ | $-0.14 \pm 0.10$ | $-0.04 \pm 0.05$ |

of the HOx-driven loss cycle due to the water vapor anomaly (Randel et al., 2024; Fleming et al., 2024). Ultimately we speculate that the stronger sAOD anomaly in WACCM6-MAM, counterbalanced by the stronger water vapor and ozone anomaly in MIROC-CHASER, is the cause of the matched forcing observed in Fig. 4a.

In Fig. 4 we also provide a comparison with available observations, with more in depth comparisons also provided in Zhuo et al. (2025). In particular we use the Global Space-based Stratospheric Aerosol Climatology version 2.22 (GloSSAC, NASA/LARC/SD/ASDC) for zonal-mean monthly-mean stratospheric aerosol optical depth and the Stratospheric Water and OzOne Satellite Homogenized dataset version 2.6 (SWOOSH,  Davis et al., 2016) for H$_2$O. In general, both models show good qualitative agreement with observations: while GloSSAC does not see the high peak at the beginning of the eruption, transport
towards the SH happens on the same timescales as the simulated one, with WACCM6-MAM showing a better match in terms of residual aerosol plume at around 60°S. In terms of H$_2$O, models generally reproduce the upward transport pattern; however, SWOOSH consistently reports higher values. As also noted in (Zhuo et al., 2025), the comparison between models and observations is primarily aimed at assessing transport patterns, since anomalies are derived differently in each case, requiring careful consideration for a meaningful quantitative comparisons. In the following section, we will discuss how the results for
WACCM6-MAM compare between fully coupled and nudged simulations.

### 3.3   Further exploration of radiative forcings in WACCM6-MAM

In this section we show a comparison of radiative forcings in the three different experiments, which only WACCM6-MAM conducted in full. This comparison provides useful insights to the different ways to define the radiative impacts of the plume
between RF, ERF and IRF.

Fig. 5 shows, across all experiments, the simulated clear-sky radiative forcing is negative and locally statistically significant in the Southern Hemisphere during the first two years after the eruption. The sign, as well as the spatial and temporal evolution, is consistent across the different model configurations and atmospheric levels considered. The differences in stratospheric
aerosol optical depth change (sAOD, Fig. 6 a-c) among the experiments are negligible: as discussed in past works, this indicate that atmospheric nudging does not necessarily improve the residual stratospheric circulation (Chrysanthou et al., 2019).





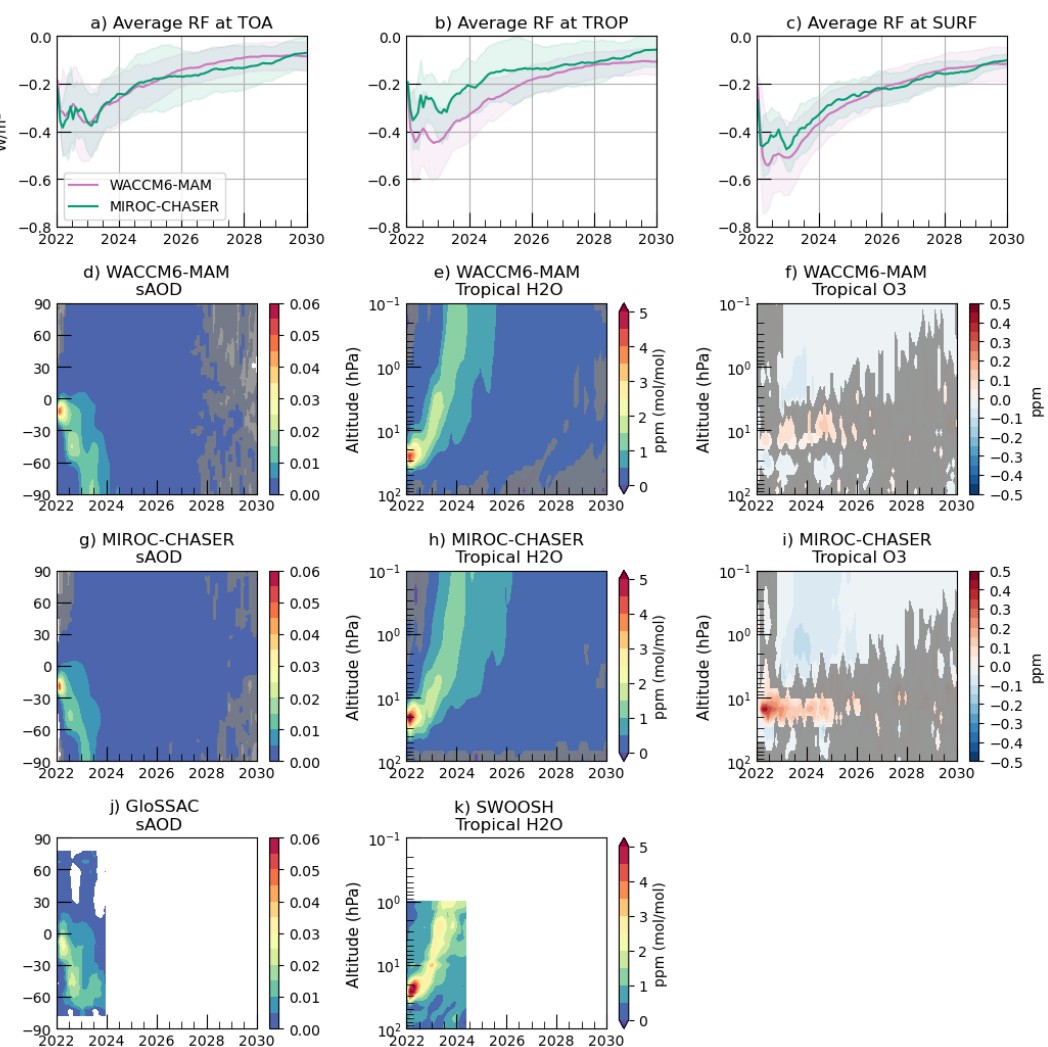

**Figure 4.** a-c) Timeseries of moving average of effective radiative forcing under clear-sky conditions, averaged between 60°S and the equator, from the free running simulatons with fixed climatological SSTs and sea ice (Fixed-SST). ERF is shown at TOA, TROP, and SURF for WACCM6-MAM and MIROC-CHASER. Solid lines represent the ensemble mean; shaded areas represent 1 standard deviation across ensemble members. Zonal mean of changes in stratospheric aerosol optical depth (sAOD) for WACCM6-MAM, MIROC-CHASER and GloSSAC (d, g, j), tropical (30 °S-N) $H_2O$ mixing ratio for WACCM6-MAM, MIROC-CHASER and SWOOSH (e, h, k) and $O_3$ mixing ratio for WACCM6-MAM and MIROC-CHASER (f and i). Changes are calculated as the difference between perturbed and unperturbed simulations for the models, and relative to the 2005–2021 climatology period for the observations. Gray areas indicate regions where the differences are not statistically significant at the 5% level based on Student's t-test.



However when including the atmospheric temperature adjustments (as is the case in the free running simulations), the radiative forcing from aerosols and water vapor is markedly smaller than in the nudged simulation. The clear-sky IRF at TOA averaged over the southern hemisphere (60°S to the equator) during the first 2 years after the eruption is -0.43 W/m$^2$ in the nudged simulations, whereas in the free running experiments it is -0.27 W/m$^2$ in both the coupled ocean and atmosphere-only cases. Despite being smaller than in the nudged simulations, such a response is still outside of natural variability, which we calculate in Fig. 7 as one standard deviation in the control ensemble (NoVolc).

When distinguishing the contribution from aerosol- and gas- radiation interaction in Fig. 8, the negative radiative forcing at TOA in the first 2 years results from the negative contribution from Hunga aerosols, and a residual positive contribution from changes in gases (contrasting with the negative gas-radiation interaction forcing seen in the nudged simulations in Fig. 3). Notably, in the free running simulations the gas contribution to radiative forcing is significantly affected by natural variability (red line in Fig. 8), in particular for all-sky RF, which hinders a confident determination of the RF response without the use of a large ensembles (Fig. A8). Moreover, in the second half of 2025, as the sAOD in the tropics returns to background levels, only the coupled experiment exhibits a significant negative radiative forcing in the Tropics at both the TOA and TROP, reaching magnitudes comparable to those observed during the first year (Fig. 7a and b), while the forcing at the SURF becomes positive. Such radiative forcing change in 2025 in the coupled experiment is primarily driven by gas-radiation interactions, with significant negative clear-sky values seen in 2025 (Figure 8c).

Changes in gas radiative interactions can be important contribution to the overall RF, and these are the result of not just the Hunga induced changes in water vapor (which are similar across the three cases), but also changes in stratospheric temperatures and ozone, as well as any associated changes in dynamics (as these modulate the temperatures and ozone, in particular in the lower stratosphere). By definition, the nudged simulation do not include any changes in temperatures or dynamics. While free running simulations include these components, the fully coupled simulation includes additional forcing from changes in the ocean variability. Differences in the gas radiative interactions and gas RF between the nudged and coupled simulations could be partly explained by the associated changes in stratospheric temperatures and ozone. Zhuo et al. (2025) has shown an upper-level cooling due to the water vapor and a lower stratospheric level warming from LW absorption from the sulfate aerosols. The decrease in upper stratospheric temperatures in the free running simulations (Fig. 6j and Fig. A9d) caused by higher water vapor at the same level, likely results in an initial positive RF at the TOA. Furthermore, the fully coupled simulations show a distinct pattern of increasing tropical lower stratospheric ozone in the first two years and decreasing ozone in 2025. Since ozone in the lower stratosphere acts as a greenhouse gas, this contributes to the positive RF in the first two years and to a negative RF in 2025 (Fig. 6g and Fig. A9g). The close correlation of changes in ozone and temperatures in this region is strongly indicative of their dynamical origin, suggesting an associated decrease in tropical upwelling in 2022-2023 and increase in 2025. As discussed in (Bednarz et al., 2025), the coupled ocean WACCM6-MAM simulations shows a significant modulation of the ENSO variability by the eruption, with La-Nina like response in 2022-2023 and an El-Nino like response in 2025. In general,





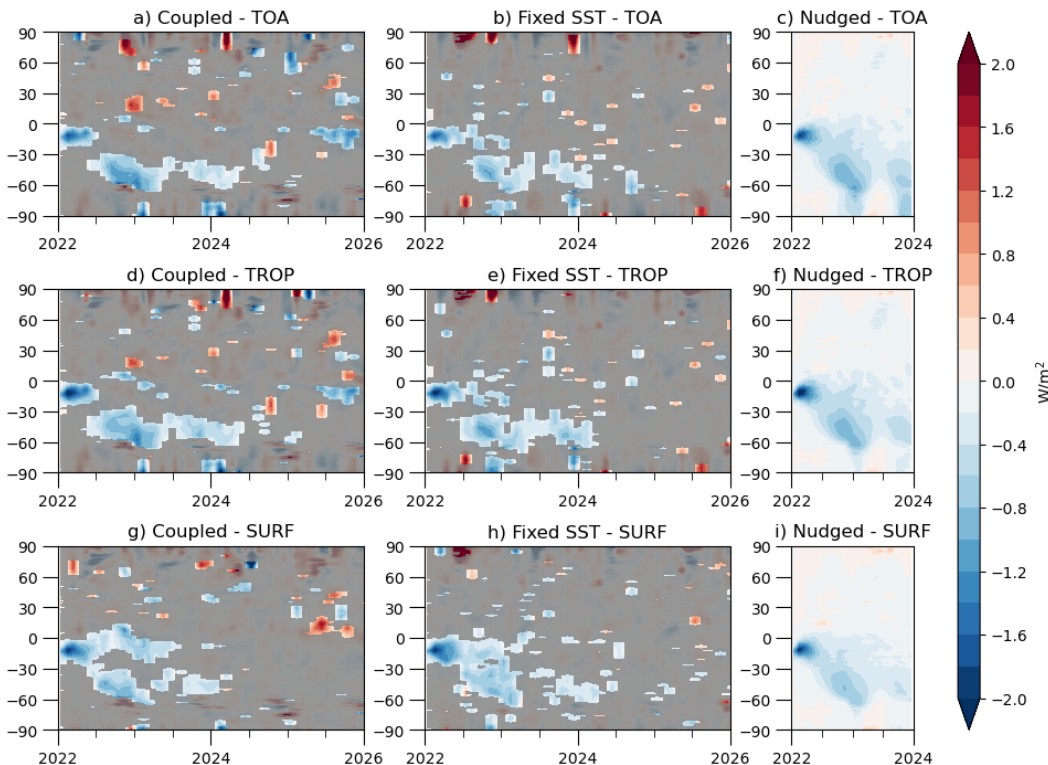

**Figure 5.** Time series of zonal mean radiative forcing under clear-sky conditions for SO2andH2O experiment in WACCM6-MAM, at TOA (a-c), TROP (d-f), and SURF (g-i). Each column represents a different model setup: first and second columns are free running experiments with either coupled ocean or climatological sea surface temperatures and sea ice (Coupled and Fixed-SST, respectively), third column indicates nudged simulations (Nudged). Results from Fixed-SST and Coupled are ensemble means over all 30 members. Gray areas indicate regions where the differences are not statistically significant at the 5% level based on Student's t-test.

ENSO is an important driver of interannual variability in tropical upwelling, which modulates lower stratospheric temperatures and ozone (Randel et al., 2009), and can therefore exert an important influence on the overall RF.

In general, while most of the results presented here are for the SH only, when considering global mean the values are reduced by approximately half, reaching a value of -0.09 W/m$^2$ at the TOA for the Coupled experiment in 2022-2023, which falls within the range of natural variability (see Fig. A7 and Table 3), and corresponds to an average sAOD of 0.005 over the same period.

## 4 Conclusions

In this work we have provided the first multi-model analyses of the radiative impacts of the Hunga Tonga-Hunga Ha'apai volcanic eruption, which co-injected large amounts of water vapor and a small amount of SO$_2$ into the stratosphere. Our multi-





**Table 3.** Radiative forcing (in W/m$^2$) at the Top of Atmosphere under clear-sky conditions in WACCM6-MAM. Forcing is averaged between 60°S and the equator and calculated over 2022-2023.

|  | Coupled (60S/Eq) | Coupled (global) | Fixed-SST (60S/Eq) | Fixed-SST (global) | Nudged (60S/Eq) | Nudged (global) |
|---|---|---|---|---|---|---|
| TOA | -0.27 ± 0.24 | -0.09 ± 0.16 | -0.27 ± 0.11 | -0.16 ± 0.09 | -0.43 | -0.24 |
| TROP | -0.37 ± 0.18 | -0.15 ± 0.09 | -0.34 ± 0.11 | -0.20 ± 0.09 | -0.44 | -0.25 |
| SURF | -0.36 ± 0.13 | -0.19 ± 0.09 | -0.37 ± 0.10 | -0.23 ± 0.09 | -0.37 | -0.20 |

model results confirm previous analyses from Zhu et al. (2022) and Stenchikov et al. (2025) which indicated a potential net negative forcing from the volcanic plume, due to the formation of a persistent layer of stratospheric sulfate aerosol whose sizes were larger (and thus exerted a stronger cooling effect) than expected based on past explosive eruptions, due to microphysical growth driven by the unusually large amounts of water vapor.

In particular, our analysis indicates a global mean effective radiative forcing at the top of the atmosphere of -0.14 ± 0.10 W/m$^2$, based on the multi-model mean from free-running simulations with fixed sea surface temperatures. If the averaging is restricted to the Southern Hemisphere, these values nearly double, highlighting the hemispheric asymmetry in the distribution of the aerosol plume and its radiative impact. When coupled with the ocean, the global forcing is smaller and more noisy (-0.09 ± 0.16 W/m$^2$), due to the eruption's significant modulation of ENSO variability, triggering a La Niña–like response in 2022–2023 and an El Niño–like response in 2025 (Bednarz et al., 2025), which, in turn, impacts tropical upwelling and alters lower stratospheric ozone in the tropics. However, the predominant effect arises from the sulfate aerosols (-0.18 ± 0.02 W/m$^2$, IRF from the multi-model mean in the Nudged SO2only) and only a marginal contribution from the water vapor. The two methods used to estimate the IRF from aerosol-radiation and gas-radiation interactions, one simulating separate injections of SO$_2$ and water vapor, and the other employing a double radiation call for their co-injection, reveal biases in IRF calculations. Our results provide useful insight that can be used to inform future climate assessments (Forster et al., 2025) that aim to identify the contributions of the single natural and anthropogenic factors to global radiative imbalance and temperatures.

It is important to note that in the simulations analyzed in this work, a value of 0.5 Tg of SO$_2$ was used (Zhu et al., 2024). Analyses by Sellitto et al. (2024), using the Infrared Atmospheric Sounding Interferometer (IASI), however, suggest that the overall SO$_2$ burden from the Hunga eruption was larger than what Carn et al. (2022) estimated using UV measurements from the Ozone Monitoring Instrument (OMI) on NASA's Aura satellite, with a lower limit of 1.0 Tg. This suggests that the results we presented here might be underestimated, if higher retrieval estimates for the sulfate aerosol burden were confirmed. In general, aerosol optical depth (AOD) is often assumed to scale linearly with radiative forcing, especially in the stratosphere, meaning that a doubling of AOD would typically result in a doubling of negative radiative forcing. However, this relationship depends not only on the total aerosol burden but also on the particle size distribution. A larger burden does not necessarily lead



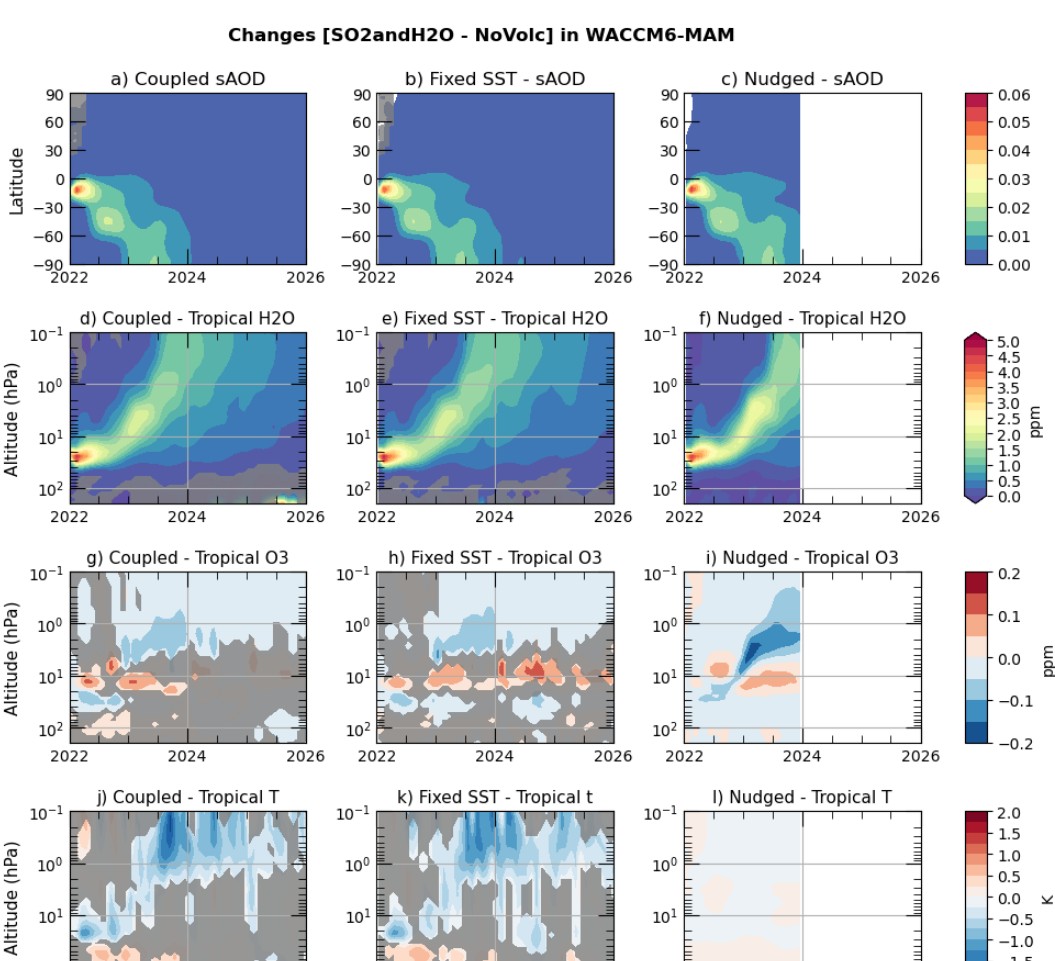

**Figure 6.** Time series of zonal mean changes in stratospheric aerosol optical depth (sAOD; panels a–c), tropical stratospheric $H_2O$ concentrations ($30°S–30°N$; panels d–f, in ppm), tropical stratospheric $O_3$ concentrations ($30°S–30°N$; panels g–i, in ppm), and tropical stratospheric temperature($30°S–30°N$; panels j–l, in K). Changes are computed as the difference between the SO2andH2O and NoVolc experiments for the (a, d, g, j) Coupled, (b, e, h, k) Fixed-SST, and (c, f, i, l) Nudged configurations. Gray areas indicate regions where the differences are not statistically significant at the 5% level based on Student's t-test.

to a proportionally larger AOD, as the AOD is highly sensitive to particle size. However, our analyses also show that the water vapor contribution to the sulfate forcing is significant, suggesting that even a doubling of the burden might not necessarily have resulted in a doubling of the negative forcing. Furthermore, our analysis shows that stratospheric water vapor significantly contributes to the initial growth of sulfate aerosols. Results from nudged WACCM6-MAM simulations indicate that, during the first six months following the eruption, the aerosol–radiation interaction IRF in the Southern Hemisphere would be over 10%

smaller (-0.34 W/m$^2$ vs -0.39 W/m$^2$) if the co-injection of water vapor is not included. However, even if the aerosol burden



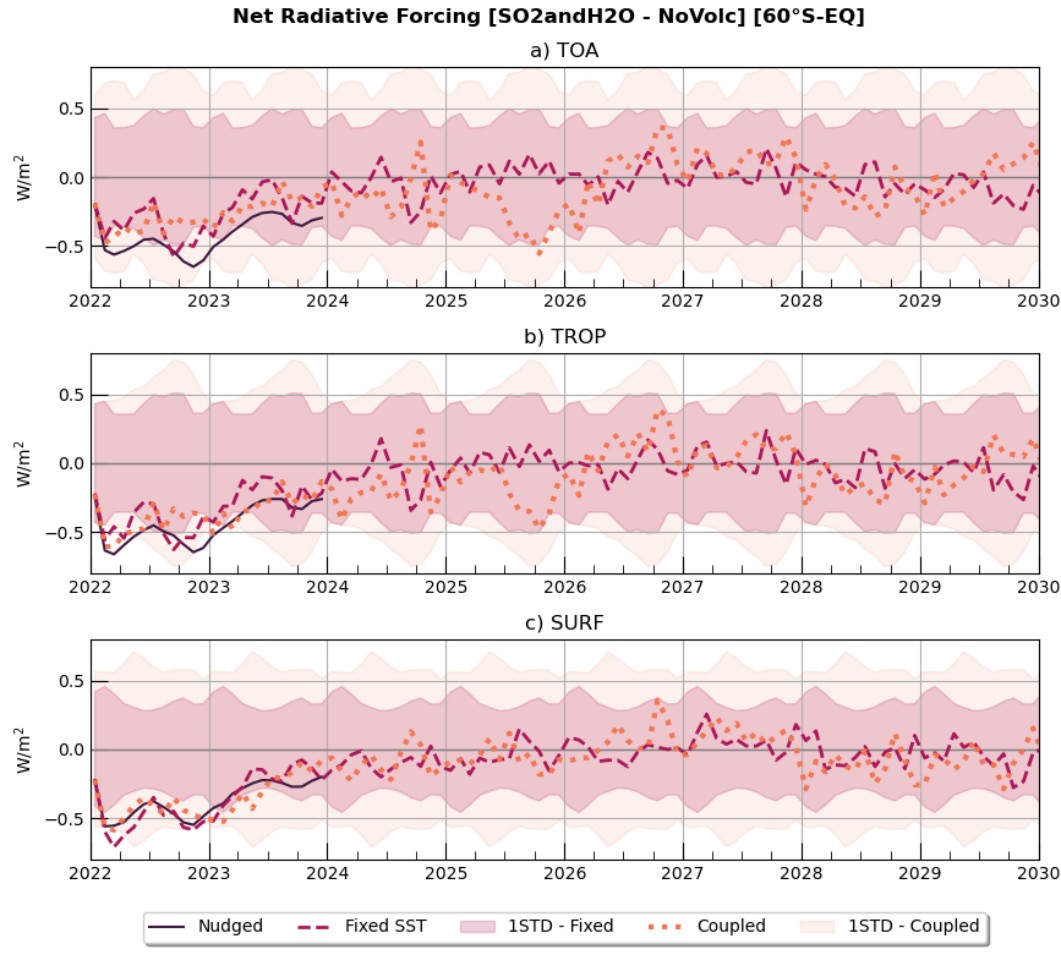

**Figure 7.** Time series of net radiative forcing under clear-sky conditions between 60°S and the equator for SO2andH2O experiment in WACCM6-MAM at TOA (a), TROP (b), and SURF (c). Solid lines indicate the results form nudged simulations, dashed lines from the free running simulations with fixed climatological SSTs and sea ice and dotted lines from the coupled experiment. Shading indicates the monthly internal variability, calculated as 1 standard deviation over 10-year simulations using a 30-member ensemble from the unperturbed NoVolc experiment.

were doubled, non-linearities in the AOD–forcing relationship would likely be less than double. Ultimately, the higher particle scattering efficiency and longer atmospheric lifetime following the Hunga eruption - attributed to its higher injection altitude and increased co-emission of water vapor (Li et al., 2024) - helps explain why an $SO_2$ injection at least 20 times smaller than Pinatubo (10-12 Tg; Ukhov et al., 2023) resulted in an sAOD only about 10 times lower and thus capable of having a small but nonetheless non-negligible contribution on the overall Earth's radiative balance over the following year.


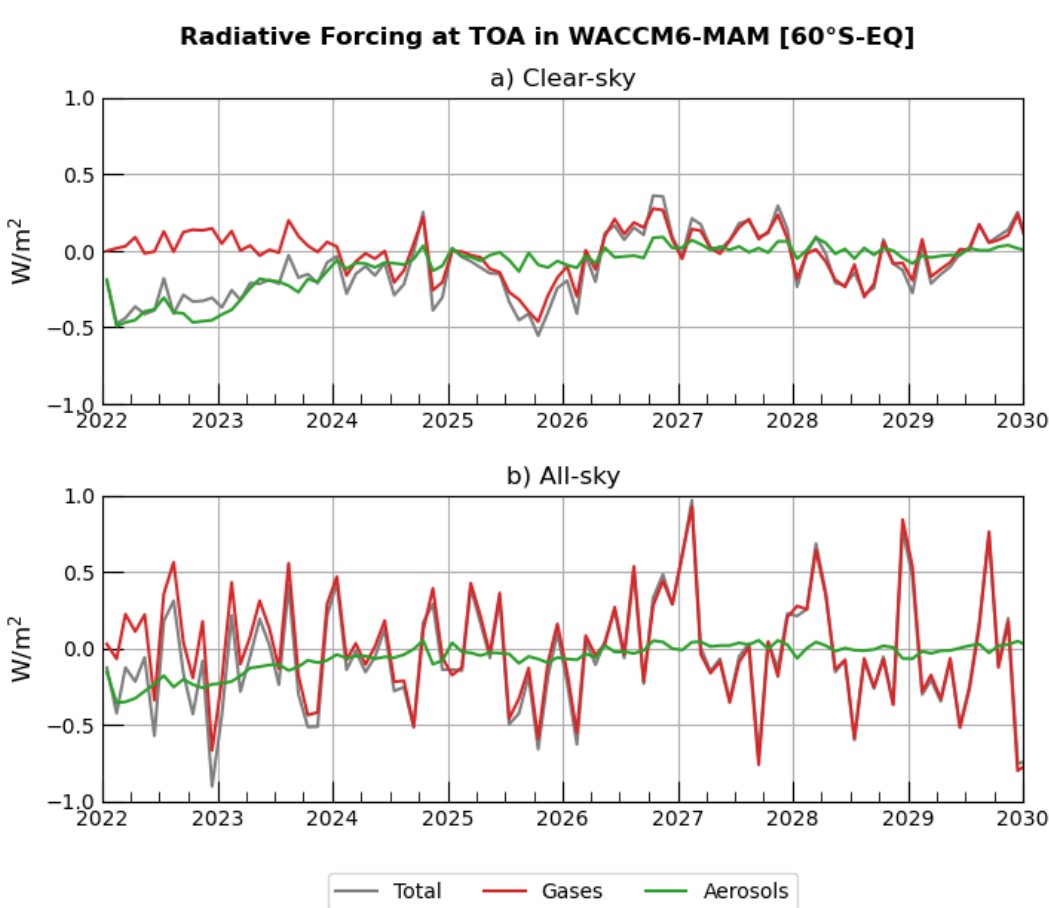

**Figure 8.** Time series of net radiative forcing at TOA for the SO2andH2O fully coupled experiment in WACCM6-MAM. RF is averaged between 60°S and the equator and calculated in Clear-sky and All-Sky conditions (a-b, respectively). Different lines represent the total RF and each contribution from aerosols only and gases only.





## Appendix A: Figures and tables

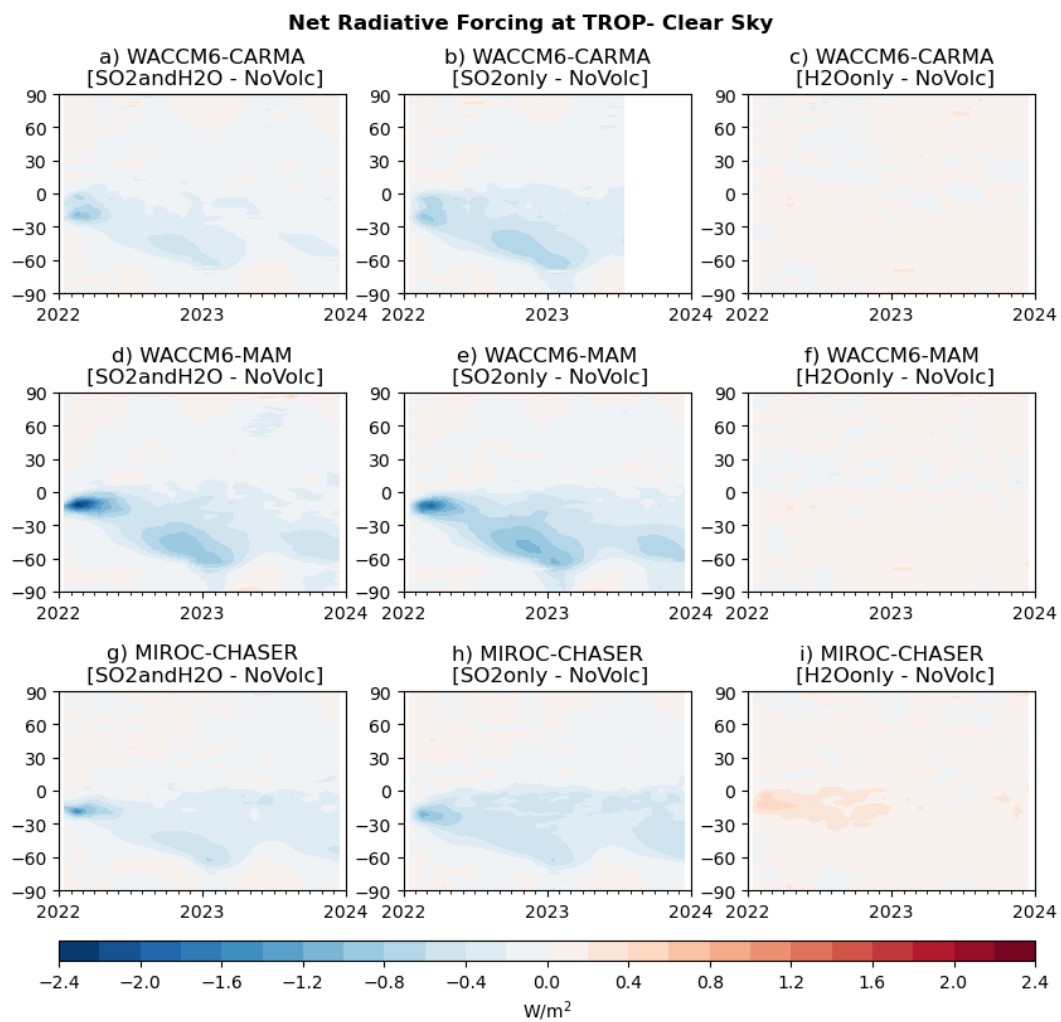

**Figure A1.** Time series of zonal mean radiative forcing at the tropopause (TROP) under clear-sky conditions from three models: WACCM6-CARMA (a–c), WACCM6-MAM (d–f), and MIROC-CHASER (g–i). Each column corresponds to a different perturbation scenario from the nudged experiment: the first column shows SO2andH2O, the second column SO2only, and the third column H2Oonly.





**Figure A2.** Time series of zonal mean radiative forcing at the surface (SURF)under clear-sky conditions from three models: WACCM6-MAM (a–c), WACCM6-CARMA (d–f), MIROC-CHASER (g–i), CAM5-CARMA (j-l), and UKESM (m). Each column corresponds to a different perturbation scenario from the nudged experiment: the first column shows SO2andH2O, the second column SO2only, and the third column H2Oonly.



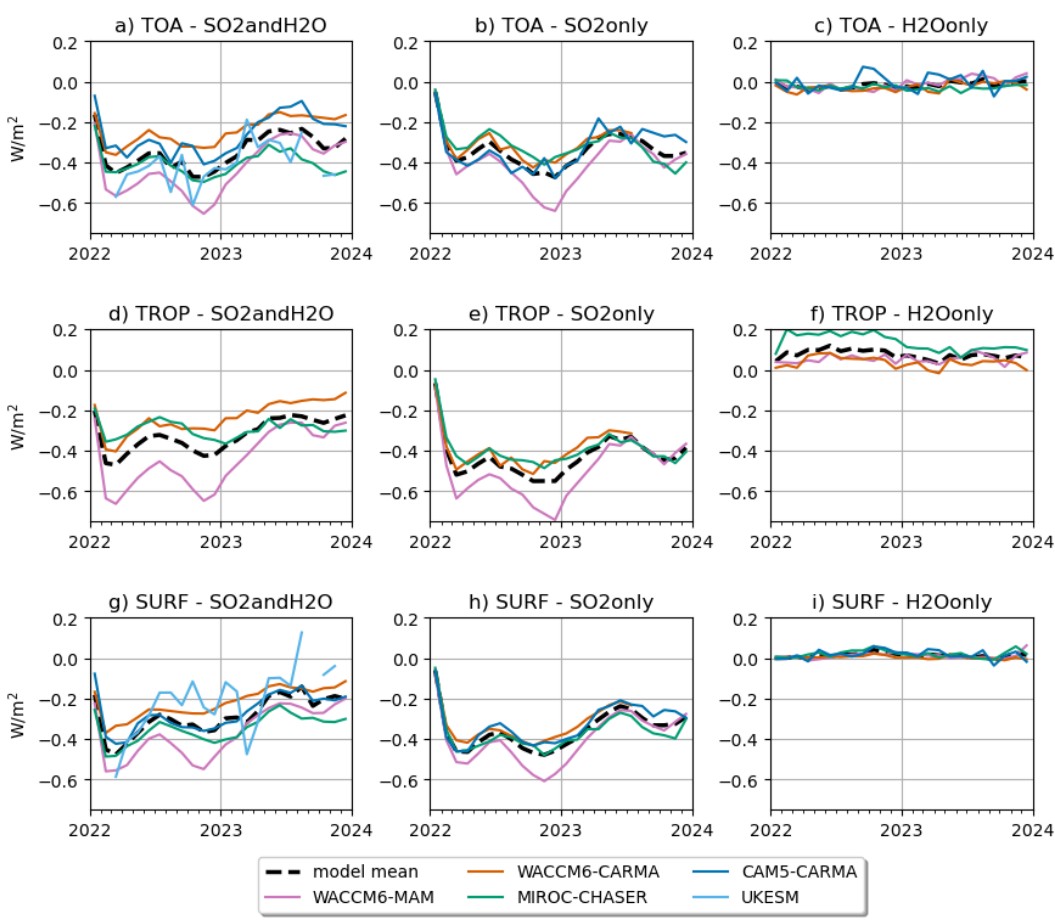

**Figure A3.** Time series of net radiative forcing under clear-sky conditions at three atmospheric levels: TOA (a, d, g), TROP (b, e, h), and SURF (c, f, i). Each column corresponds to a different perturbation scenario from the nudged experiment: the first column shows SO2andH2O, the second column SO2only, and the third column H2Oonly. The black dashed line represents the multi-model mean, each models is represented with a different color.



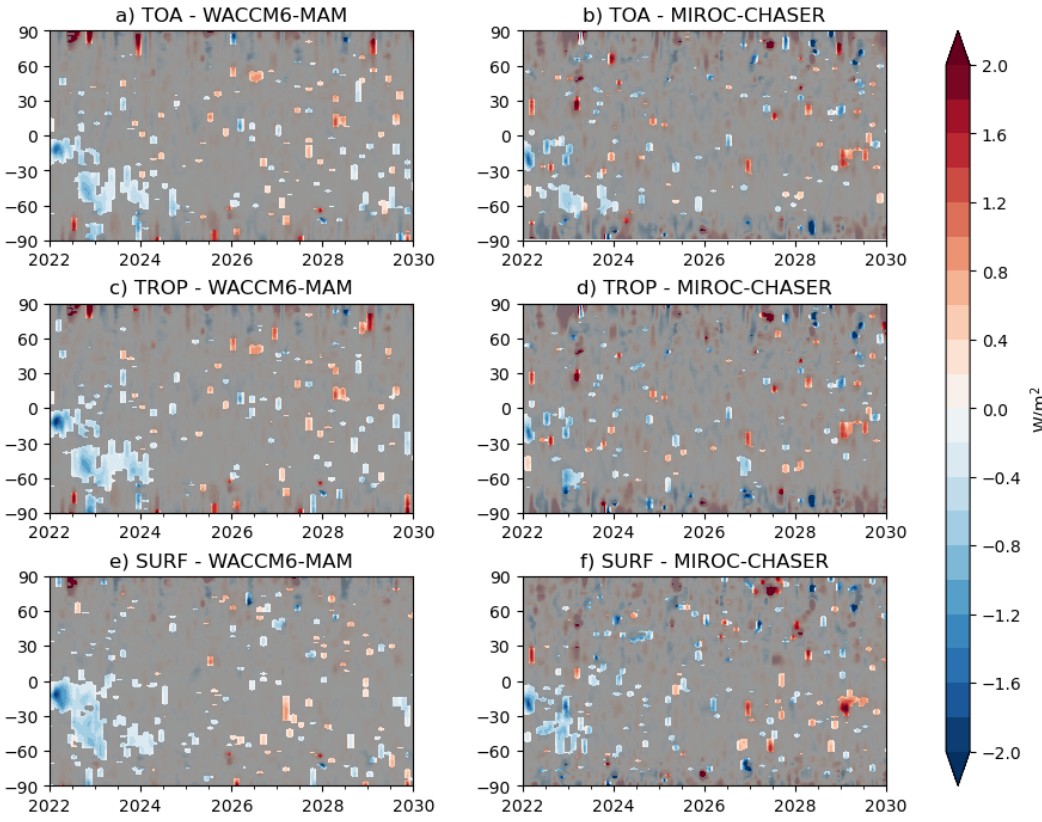

**Figure A4.** Time series of zonal mean radiative forcing under clear-sky conditions for the SO2andH2O experiment with fixed SST (Fixed-SST). RF is calculated at TOA (a and b), TROP (b and c), and SURF (e and f) in WACCM6-MAM (first column) and MIROC-CHASER (second column). Gray areas indicate regions where the differences are not statistically significant at the 5% level based on Student's t-test.



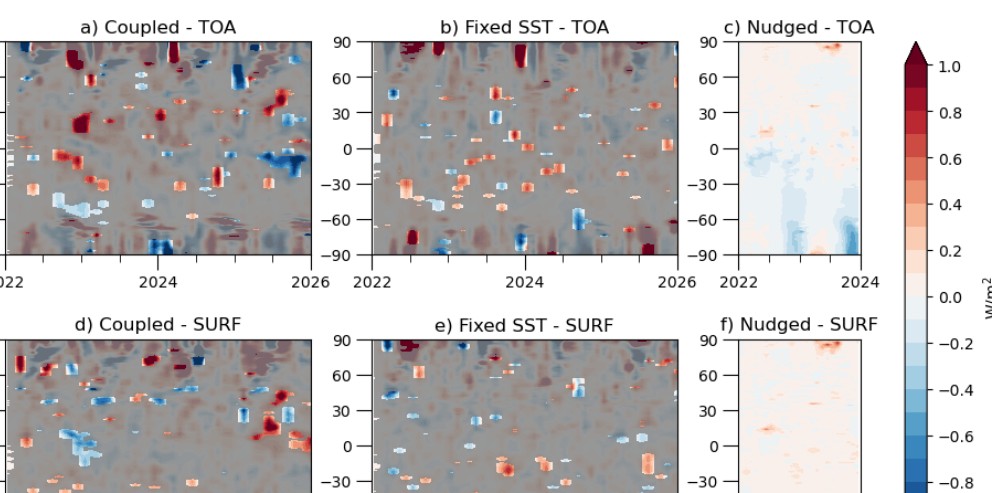

**Figure A5.** Time series of zonal mean gas-radiation interaction RF under clear-sky conditions for the SO2andH2O experiment, at TOA (a-c), and SURF (d-f). Each column represents a different model setup: first and second columns are free running experiments with either coupled ocean or climatological sea surface temperatures and sea ice (Coupled and Fixed-SST, respectively), third column indicates nudged simulations (Nudged). Results from Fixed-SST and Coupled are ensemble means. Gray areas indicate regions where the differences are not statistically significant at the 5% level based on Student's t-test.

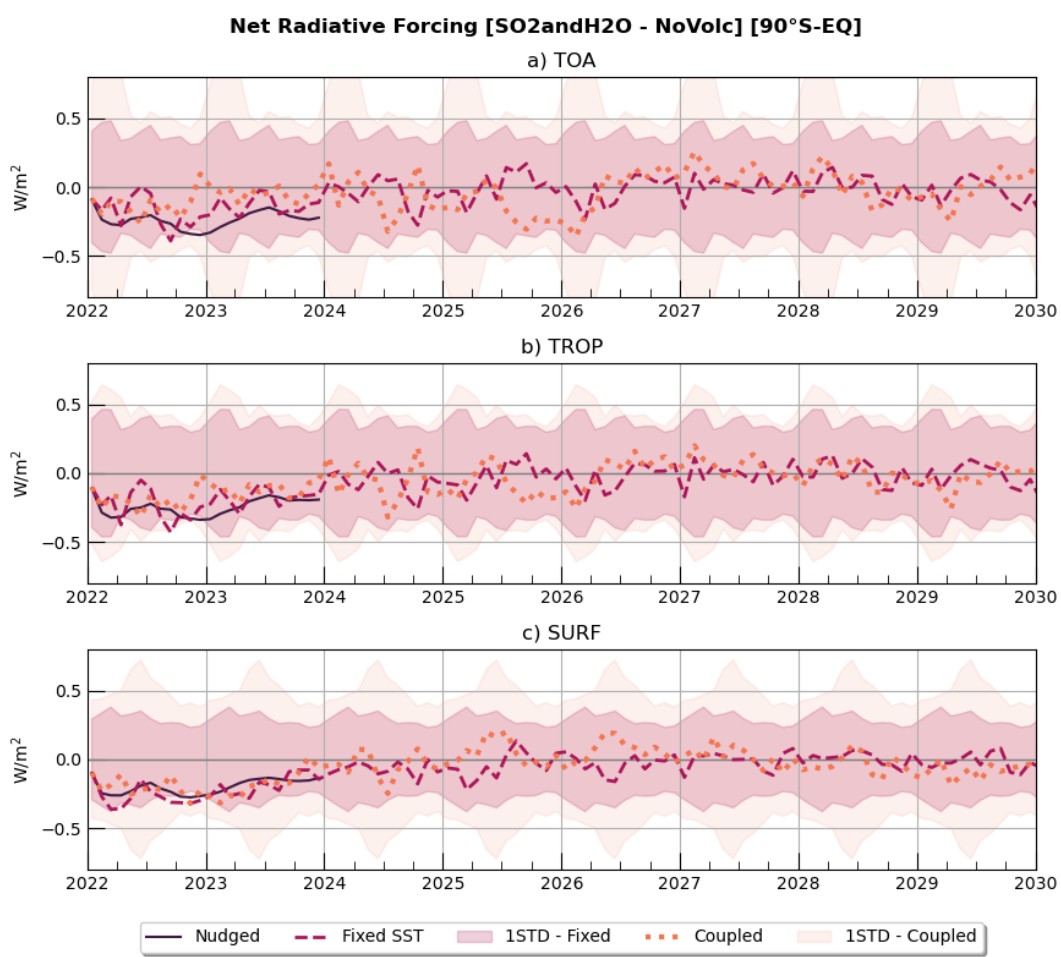

**Figure A6.** Time series of global net radiative forcing under clear-sky conditions for SO2andH2O experiment in WACCM6-MAM at TOA (a), TROP (b), and SURF (c). Solid lines indicate the results form nudged simulation (Nudged), dashed lines from the fixed SST simulation (Fixed-SST) and dotted lines from the coupled experiment (Coupled). Shading indicates the monthly internal variability, calculated as 1 standard deviation over 10-year simulations using a 30-member ensemble from the unperturbed NoVolc experiment.

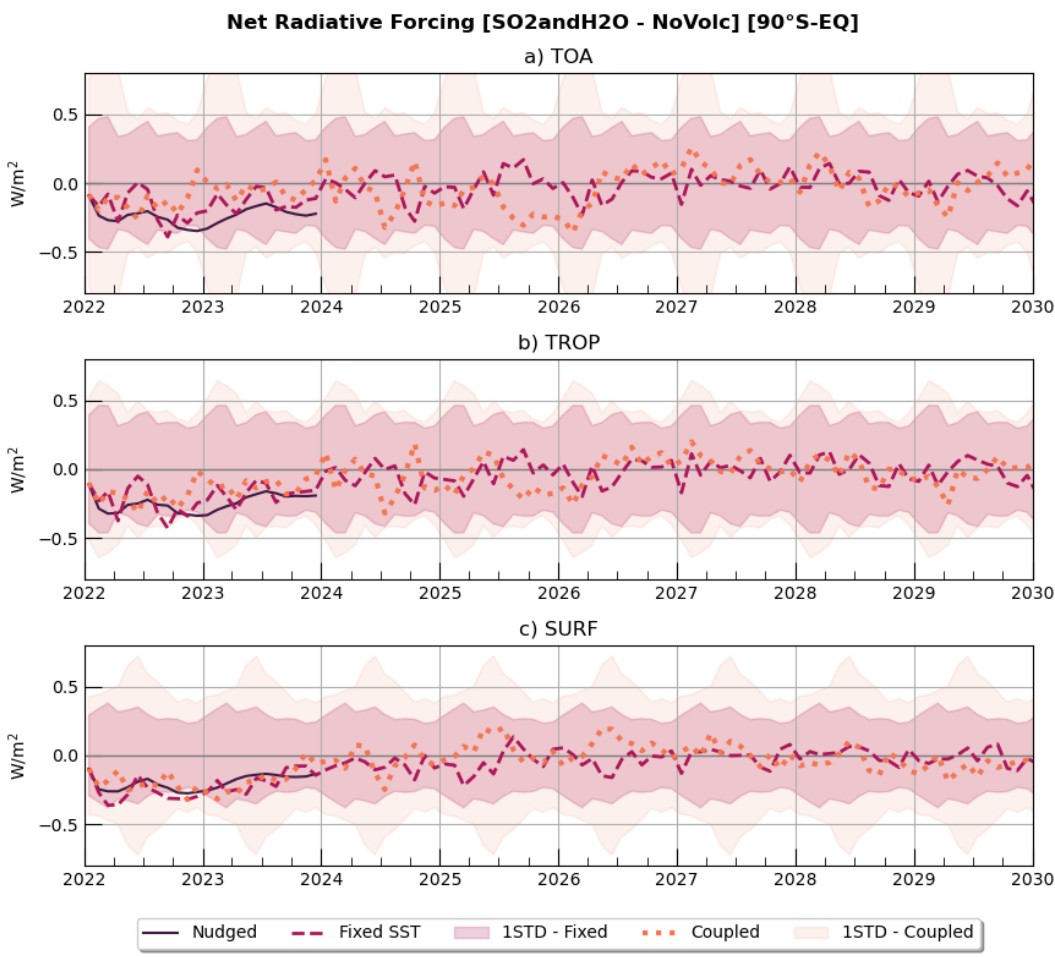

**Figure A7.** Time series of global net radiative forcing under clear-sky conditions for SO2andH2O experiment in WACCM6-MAM at TOA (a), TROP (b), and SURF (c). Solid lines indicate the results form nudged simulation (Nudged), dashed lines from the fixed SST simulation (Fixed-SST) and dotted lines from the coupled experiment (Coupled). Shading indicates the monthly internal variability, calculated as 1 standard deviation over 10-year simulations using a 30-member ensemble from the unperturbed NoVolc experiment.





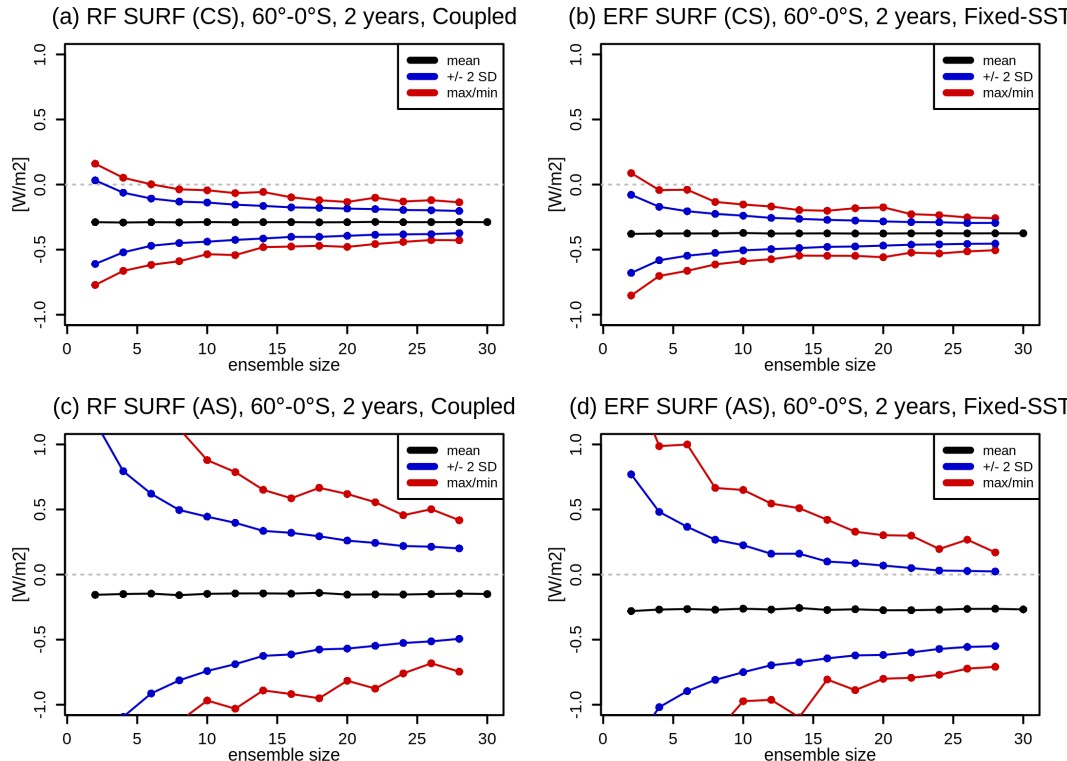

**Figure A8.** Detectability of the 2022-2023 average RF SURF response in the Coupled simulations (left) and the ERF response in the Fixed-SST simulations (right) for clear-sky (top) and all-sky (bottom) conditions. Results obtained by randomly subsampling each ensemble with replacement to obtain 2000 artificial ensembles each of different ensemble size. Black lines denote the mean response, and blue and red lines indicate the $\pm 2$ standard deviation and the max/min ranges, respectively, of the possible responses. The results show that a 2-year mean clear-sky RF/ERF response is detectable already with a few ensemble members ($\sim 5$) but requires more members to constrain the magnitude with any confidence. In contrast, the uncertainty of the all-sky response is substantially larger, particularly in the Coupled experiment but also in the Fixed-SST experiment, where even 30 ensemble members may be insufficient to determine even the sign of the response with any confidence.





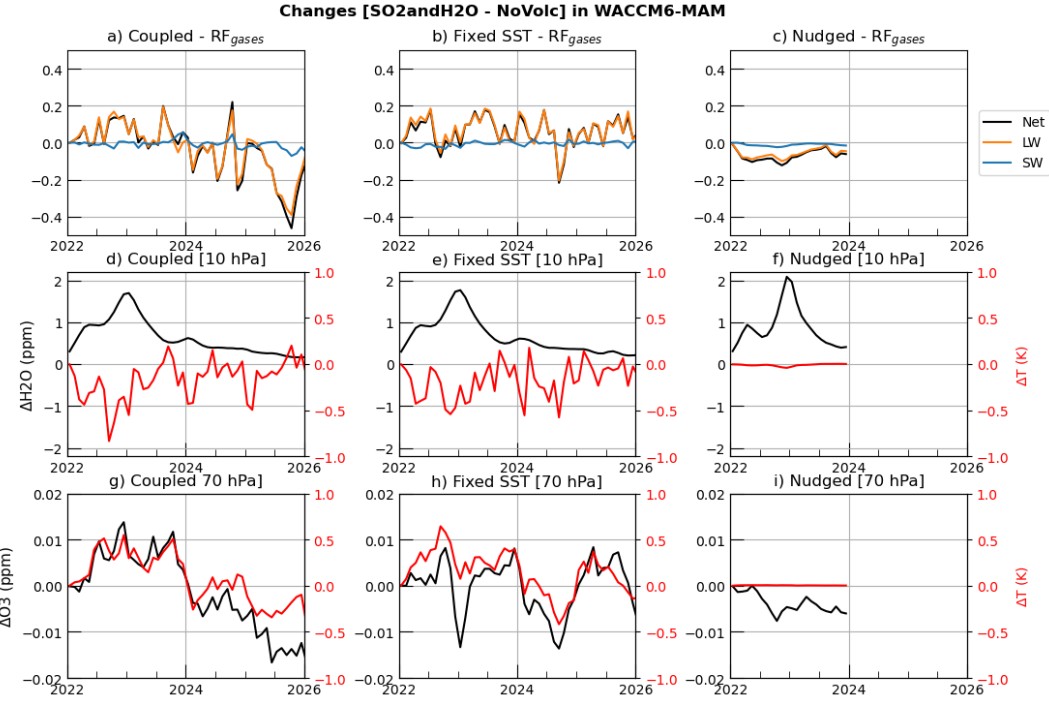

**Figure A9.** Time series of net, shortwave (SW), and longwave (LW) gas-radiation interaction RF (sAOD; panels a–c), tropical $H_2O$ concentrations and tropical temperature at 10 hPa ($30°S–30°N$; panels d–f, in ppm and K, respectively), and tropical $O_3$ concentrations and tropical temperature at 70 hPa ($30°S–30°N$; panels g–i, in ppm and K, respectively). Changes are computed as the difference between the SO2andH2O and NoVolc experiments for the (a, d, g) Coupled, (b, e, h) Fixed-SST, and (c, f, i) Nudged configurations.

**Table A1.** Global effective radiative forcing (in W/m$^2$) at the Top of Atmosphere under clear-sky conditions.

| Model | 6 Months | 1 Year | 2 Years | 5 Years | 10 Years |
|---|---|---|---|---|---|
| WACCM6-MAM | $-0.11 \pm 0.15$ | $-0.18 \pm 0.13$ | $-0.16 \pm 0.09$ | $-0.07 \pm 0.07$ | $-0.03 \pm 0.06$ |
| MIROC-CHASER | $-0.08 \pm 0.17$ | $-0.15 \pm 0.12$ | $-0.11 \pm 0.10$ | $-0.10 \pm 0.08$ | $-0.04 \pm 0.06$ |





**Table A2.** Effective radiative forcing (in W/m$^2$) at the Tropopause under clear-sky conditions. Forcing is averaged between 60°S and the equator.

| Model | 6 Months | 1 Year | 2 Years | 5 Years | 10 Years |
|---|---|---|---|---|---|
| WACCM6-MAM | -0.40 ± 0.14 | -0.44 ± 0.15 | -0.34 ± 0.11 | -0.15 ± 0.06 | -0.09 ± 0.05 |
| MIROC-CHASER | -0.25 ± 0.15 | -0.31 ± 0.12 | -0.21 ± 0.12 | -0.12 ± 0.03 | -0.04 ± 0.05 |





**Table A3.** Effective radiative forcing (in W/m$^2$) at the Surface under clear-sky conditions. Forcing is averaged between 60°S and the equator.

| Model | 6 Months | 1 Year | 2 Years | 5 Years | 10 Years |
|---|---|---|---|---|---|
| WACCM6-MAM | $-0.53 \pm 0.16$ | $-0.51 \pm 0.16$ | $-0.37 \pm 0.10$ | $-0.18 \pm 0.07$ | $-0.09 \pm 0.07$ |
| MIROC-CHASER | $-0.42 \pm 0.12$ | $-0.48 \pm 0.09$ | $-0.33 \pm 0.08$ | $-0.19 \pm 0.06$ | $-0.08 \pm 0.03$ |



*Data availability.* The HTHH-MOC Project teams are working to make the data available publicly. We expect this to happen before final the publication of this manuscript.

*Author contributions.* IQ performed all analyses and wrote the manuscript with DV and EB. XW, EB, JZ, WY performed the WACCM6-MAM simulations. YZ designed the simulations. GS assisted with the analyses and contributed to the discussion of the results. C-CL performed the WACCM6-CARMA simulations with ST. GM performed the UKESM simulations. YP and PY performed the CAM5-CARMA simulations. SW conducted MIROC-CHASER simulations, postprocessed and uploaded the model data on JASMIN, under supervision of TS, who developed the model aerosol microphysics scheme. All authors contributed to the revision of the manuscript.

*Competing interests.* The authors declare no competing interest. Some authors are editorial board members of the journal ACP.

*Acknowledgements.* NCAR's Community Earth System Model project is supported primarily by the National Science Foundation under Co-operative Agreement No. 1852977. Computing and data storage resources, including the Derecho supercomputer (doi:10.5065/qx9a-pg09), were provided by the Computational and Information Systems Laboratory (CISL) at NCAR. Ilaria Quaglia acknowledges support from the US Simons Foundation (grant ref. MPS-SRM-00005203). Ewa Bednarz acknowledges support from the National Oceanic and Atmospheric 330 Administration (NOAA) cooperative agreement NA22OAR4320151 and the Earth's Radiative Budget (ERB) program. Simone Tilmes acknowledges support by the NOAA Climate Program Office Earth's Radiation Budget award no. 03-01-07-001 and NA22OAR4310477. Shingo Watanabe and Takashi Sekiya were supported by MEXT-Program for the advanced studies of climate change projection (SENTAN) Grant Number JPMXD0722681344. MIROC-CHASER simulations were performed using the Earth Simulator.



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
