# Peer review of "Multi-model analysis of the radiative impacts of the 2022 Hunga eruption indicates a significant cooling contribution from the volcanic plume"

_EGUsphere, 2025_

## Referee Comment (RC1)

Review of "*Multi-model analysis of the radiative impacts of the 2022 Hunga eruption indicates a significant cooling contribution from the volcanic plume*" by Quaglia et al.,

The current manuscript makes a comprehensive analysis of the radiative impacts due to the Hunga volcanic eruption using multiple models and clearly delineates the radiative forcing due to the aerosols and gases, and also discusses its possible uncertainties in the analysis.

I highly appreciate the authors' extensive work employing multiple models to perform decadal simulations of radiative forcing from the Hunga eruption, one of the most significant volcanic events in recent history. The authors also compared the model results with the observational dataset wherever required. The manuscript is technically sound, and the results are presented concisely. However, my main concern is the discussion section, which requires further improvement and refinement. To be specific, several figures are under-discussed, which reduces the overall impact of the work despite the availability of strong results. Providing a more detailed interpretation of the model outputs in each figure would substantially enhance both the clarity and significance of the study. In addition, the novelty of the present study is not clearly argued. I have provided a few major and minor comments to assist the authors in improving the scientific arguments and presentation style of the manuscript. With these improvements, I believe the manuscript will be suitable for publication. I therefore recommend a major revision.

**Major comments**

1. In Sect. 1 (Introduction), the authors summarize previous works that estimated the radiative forcing of the Hunga eruption. However, the distinction between the present study and these earlier studies is not explicitly articulated. Specifically, while the current and previous works (Sellitto et al., 2022; Zhu et al., 2022; Jenkins et al., 2023; Schoeberl et al., 2024; Stenchikov et al., 2025) all examine the radiative impacts of the Hunga eruption, it remains unclear in which aspects this study provides a novel contribution. For example, does the current study differ in methodology, model choice, experimental setup, dataset, study period, atmospheric level of the forcing, or spatial domain? A clearer discussion of the study's novelty would better engage the reader and also highlight the significance of the present work.

2. The study has used GloSSAC for sAOD and SWOOSH for $H_2O$ to compare the model output with observation in Fig. 4. But the details about them are included very briefly in P.11 L.216–218. In addition, information about other observational datasets which are used for the simulations (if any) is missing. I recommend that authors create a dedicated data section which outlines all the observational and meteorological datasets used in the study for the reproducibility of the results and to increase the reliability of the findings.

3. One of the key arguments of this study is that the radiative forcing under the $SO_2 + H_2O$ combined condition is higher than under the $SO_2$ condition alone, and the authors have attributed this observation to enhanced sulfate aerosol growth in the presence of $H_2O$. However, this claim is not sufficiently supported with evidence in the current manuscript. This point could be strengthened by including a plot of the aerosol size distribution under the two scenarios ($SO_2 + H_2O$ and $SO_2$ alone), which would provide direct support for the stated mechanism.

4. The flow and discussions made in Sect. 3.1 and 3.2 are clear enough, but Sect. 3.3 is confusing and very difficult to follow. The primary reason for the confusion is the authors' abrupt shift to referencing multiple figures. This shift complicates the clarity of their argument. For example, the authors begin discussing Fig. 5 but abruptly shift to Fig. 6 without completing the explanation of Fig. 5. This pattern repeats throughout the section, creating a disorganized flow. Nearly seven figures are cited here, yet none of them are discussed in sufficient detail. I would recommend that authors reorganize this section by discussing each figure with relevant interpretations and scientific arguments, and move on to another figure. I would also suggest that the authors change the title of Sect. 3.3 to make it more descriptive.

5. Another major issue is that most of the figures are underdiscussed, i.e., in any figure, only two to four panels are discussed, leaving out the rest of the panels untouched, which questions their very purpose. Initially, I thought this specific issue was minor and restricted to Fig. 2 alone (please see minor comment no. 7 for more details). But later, I realized it was prevalent among most of the figures, making it a major issue. I highly recommend that the authors increase discussion on each figure (if not all panels, try to explain the most panels), and connect them to the objective of the study. If some figures are not worth discussing or if the authors feel that such discussion may impact the manuscript flow, they can be safely moved to the supplementary section.

**Minor comments**

1. I have a suggestion regarding the title of the manuscript, "*Multi-model analysis of the radiative impacts of the 2022 Hunga eruption indicates a significant cooling contribution from the volcanic plume.*" I appreciate that the authors have highlighted the key finding (i.e., the cooling effect of the Hunga eruption) in the title. However, since previous studies have already established that the Hunga plume resulted in a net cooling, the current title does not appear very compelling and intriguing. I believe it would be more impactful to emphasize the amplification of this cooling (net radiative forcing) due to the growth of sulfate aerosols caused by the water-enriched stratosphere. Also, the title looks lengthy at present. So, if the authors are changing the title, it is better to be concise.

2. P.2 L.36: The authors have used the phrase "net warming" to refer to positive forcing. And at L.38 and L.40, the phrase "negative forcing" is used to refer to net cooling. These lines (L.35 to L.40) highlight the contrasting findings among the previous studies, as any good introduction should. But inconsistent usage of the phrase referring to the radiative forcing decreases the readability of these sentences. Hence, I would like to suggest that the authors use either "net warming" and "net cooling" or "positive forcing" and "negative forcing" to increase readability.

3. I appreciate the authors for providing details about the types of experiments and models employed in this study in Sect. 2.1 in P.3 and about the injection parameters through Table 1. However, for a common reader like me, it takes two to three reads to get clarity and

understand the method itself. It also disrupts the flow of the manuscript. To rectify it, I highly encourage the authors to add a graphical representation of the methods or a table which clearly describes all the experiments and models concisely. In addition, even though the current manuscript is part of the HTHH-MOC project and details of the employed methods are already furnished in the main paper (Zhu et al., 2025), I believe it is important to furnish minimal information (if not extensive) about the experiment design in the current manuscript for enhanced clarity.

4. In P.5 L.109–124, a few radiative forcing (RF) formulas are mentioned. Firstly, please consider adding equation numbers for all of them. Secondly, as the authors have clearly mentioned that "*Unless otherwise specified, all values are assumed to be calculated under Clear-Sky conditions*" at L.107, it is not essential to include 'clear' as a superscript in each term. In addition, the definition of '$F_{volc}^{clear}$' and '$F_{control}^{clear}$' varies in each formula, and their description is given at the end. However, it looks a bit odd to write them in this fashion. I suggest the authors consider rewriting the equation with more detailed subscripts and superscripts, something like what is given below.

$$RF^{Coupled} = F_{volc}^{coupled,SO_2,H_2O} - F_{control}^{coupled,SO_2,H_2O} \qquad (1)$$

$$ERF^{Fixed-SST} = F_{volc}^{Fixed-SST,SO_2,H_2O} - F_{control}^{Fixed-SST,SO_2,H_2O} \qquad (2) \text{ and so on.}$$

5. In Fig. 1 (in all panels, except H2O_only cases), negative RF ($\sim$-0.4 Wm$^{-2}$) is observed at the high-latitude (60°–90°S) from November 2022 to February 2023. But thereafter, the RF becomes close to 0 Wm$^{-2}$ during April–September 2023 (Austral winter). This feature is consistent across all models and experiment setups and also appears in the Austral winter of 2022. Since the authors have clearly stated in P.6 at L.148 that IRF also considers the polar stratospheric clouds (PSCs), these RF values correspond to PSC as well as other aerosols too (as it is a clear-sky condition and not clean-sky). In this scenario, is RF $\sim$0 Wm$^{-2}$ caused by the formation of PSC during the Austral winter? Is it because net cooling by stratospheric aerosols is countered by net warming by the PSC? Please describe.

6. Please explain the cause of the data gap in Fig. 1e and m.

7. The authors have estimated the RF at three key atmospheric levels: TOA, TROP, and SURF, and done an excellent job of explaining physical implications and reasons for choosing these three levels in P5 at L.126–132. These results are shown in Fig. 2 as well, with interpretation included on P.6 at L.156–162. While the discussions of RF at TOA and TROP are briefly touched on, the SURF RF has not been addressed. If the SURF RF is not required for the discussion, then its presence becomes questionable. Moving less important plots (panels) to the appendix or supplementary section may help the readers to be focused on the main argument which authors are trying to make. In addition, the discussion could

be strengthened by elaborating on the causes of the observed variation in RF across latitude bands and their temporal evolution.

(i) To be specific, for the case of SO2andH2O, the global mean (Fig. 2a, black solid line), and 60°S–Eq mean RF at TOA (Fig. 2d) show that the strongest negative RF occurred almost at the end of 2022. In contrast, at the TROP and SURF, the strongest negative RF occurred much earlier, around February 2022. What could be the possible physical mechanism of this result?

(ii) In the high-latitude band (60–90°S), RF is close to 0 Wm$^{-2}$ during the first six months of 2022 at all three levels (TOA, TROP, and SURF; third row in Fig. 2). However, a sudden strong negative RF exceeding -0.5 W m$^{-2}$ appears in January 2023, which rapidly decays and returns to ~0 W m$^{-2}$ by the peak Austral winter of 2023. Following this, another negative RF peak occurs at the end of 2023. What physical processes drive this temporal variation, particularly the ~0 W m$^{-2}$ RF during the Austral winter? This point is directly connected to my earlier comment on Fig. 1 (see major comment no. 5). A solid discussion on these aspects could potentially strengthen the discussion and make it comprehensive.

8. The authors have written, "*The fact that the forcing becomes similar later on and shows a slight reduction in the last few months of 2023, may suggest a balance between larger particles and shorter lifetime for the two cases*" in P.7 at L.170–172. Could it also be due to the polar transport of volcanic emissions from mid-to-high latitude, as evident from Fig. 1?

9. P.7 L.185–187: This is one of the interesting and important findings made in the present study and also included in the abstract. However, without a plot or supporting data, it appears speculative. To strengthen this claim, consider adding a bar plot (or any other suitable plot authors wish) comparing aerosol sizes under SO2andH2O and SO2 experiments, for both clear-sky and all-sky scenarios. This would provide clearer evidence to support the finding.

10. P.7 L.194–195: Any possible explanation for the discrepancies between the models at the tropopause?

11. P.10 L205: What does 'cloud' refer to in this context? Are the authors referring to the aerosol plume from the Hunga eruption? If so, please use the term 'aerosol plume' for consistency, since this phrase is already used at l.204. If not, kindly clarify what is meant by 'cloud' here.

11. P.13 L.249: The Fig. A8 is cited before Figs. A1–A7 are not even mentioned. Except Fig. A7, which is cited at L.275 in P.14, Figs. A1–A6 are never cited at all. If these figures are not even needed to be cited in the main text and do not contribute to the discussion, their inclusion even in the appendix is questionable.

**Technical comments:**

1. Throughout the manuscript, the unit for the radiative forcing is written as 'W/m$^2$.' I would like to suggest changing them to 'Wm$^{-2}$' both in the main text and plots as per the ACP template.

2. P.4 L91–92: The instantaneous radiative forcing is already abbreviated as IRF at L.82. Hence, it is not necessary to abbreviate it again. Please correct.

3. P.5 P.120: Write 'instantaneous radiative forcing' as IRF.

4. P.6 L.134: Please write 'Section 3.1' as 'Sect. 3.1'

5. P.6 L.138: The first paragraph of Sect. 3 provides the outline of the result and discussion section. While doing so, it would be better if the corresponding subsections are cited at the appropriate place, as shown below.

6. "*Following that, we provide analyses of the two models which provided the 10-year long free-running simulations with prescribed climatological SSTs and sea ice **(Sect. 3.2)**: these analyses allow us to understand the long-term behavior of the forcing as well as include the combined chemical and dynamical impacts and temperature adjustments.*"

7. "*Finally, we complete those with an analysis of fully-coupled simulations in WACCM6-MAM **(Sect. 3.3)**.*"

8. P.6 L147: Replace 'timeseries' with 'time series'

9. P.7 L.165: Replace 'Section 2.2' with 'Sect. 2.2'

10. P.7 L189: Replace 'Fig.4' with 'Figure 4'

11. P.7 L189: Does the 't' in the formula refer to 'time'? Please clarify.

12. In the caption of Fig. 2, it is mentioned that the first row plots correspond to 60°S–90°N. But in the title of the panel a–c, it is written as 90°N–S. Please correct.

13. P.7 L200: The stratospheric aerosol optical depth is abbreviated as 'sAOD' at L.199 and referred again as 'stratospheric AOD' at L.200. Once an abbreviation is introduced, it should be used consistently throughout the main text (except in figure captions and the conclusion). Consistency in terminology will improve both readability and the overall flow of the manuscript.

14. In Fig. 3d and e, please make a common y-axis limit. At present, the panel d has a slightly wider ylimit than panel e.

15. In the caption of Fig. 3, replace 'Section 2.2' with 'Sect. 2.2'

16. P.11 L.222: Replace "(Zhuo et al., 2025)" with "Zhuo et al., (2025)"

17. P. 12: In Fig. 4's caption, clearly indicate what the green and magenta shading region (panels a to c) represent.

18. P.11 L.232: Replace 'Fig. 5' with 'Figure 5'

19. P.11 L.234–235: Replace 'stratospheric aerosol optical depth' with 'sAOD' as the acronym is already introduced.

20. In the caption of Fig. 4, replace 'timeseries' with 'time series' and in the first line write 'effective radiative forcing' as 'effective radiative forcing (ERF)'.

21. In Fig. 4 (e, f, h, i, and k), 'Altitude (hPa)' is given as the ylabel. Usually, altitudes are given in units of m or km. Is it supposed to be 'Pressure (hPa)'? Please check and correct.

22. P.13 L.262: What does LW mean here? Please expand it.

23. P.13 L270: Expand ENSO.

24. Figures 6, 7, and 8, and Table 3 appear in the Conclusion Sect. 4. I suggest the authors move them to Sect. 3.3, where relevant discussion using these figures and tables is made.

**References**

Jenkins, S., Smith, C., Allen, M., and Grainger, R.: Tonga eruption increases chance of temporary surface temperature anomaly above 1.5 °C, Nature Climate Change, 13, 127–129, https://doi.org/10.1038/s41558-022-01568-2, 2023.

Sellitto, P., Podglajen, A., Belhadji, R., Boichu, M., Carboni, E., Cuesta, J., Duchamp, C., Kloss, C., Siddans, R., BÚgue, N., Blarel, L., Jegou, F., Khaykin, S., Renard, J. B., and Legras, B.: The unexpected radiative impact of the Hunga Tonga eruption of 15th January 2022, Communications Earth & Environment, 3, 288, https://doi.org/10.1038/s43247-022-00618-z, 2022.

Schoeberl, M. R., Wang, Y., Taha, G., Zawada, D. J., Ueyama, R., and Dessler, A.: Evolution of the Climate Forcing During the Two Years After the Hunga Tonga-Hunga Ha'apai Eruption, Journal of Geophysical Research: Atmospheres, 129, e2024JD041 296, https://doi.org/https://doi.org/10.1029/2024JD041296, e2024JD041296 2024JD041296, 2024.

Stenchikov, G., Ukhov, A., and Osipov, S.: Modeling the Radiative Forcing and Atmospheric Temperature Perturbations Caused by the 2022 Hunga Volcano Explosion, Journal of Geophysical Research: Atmospheres, 130, e2024JD041 940, https://doi.org/https://doi.org/10.1029/2024JD041940, e2024JD041940 2024JD041940, 2025.

Zhu, Y., Bardeen, C. G., Tilmes, S., Mills, M. J., Wang, X., Harvey, V. L., Taha, G., Kinnison, D., Portmann, R. W., Yu, P., Rosenlof, K. H., Avery, M., Kloss, C., Li, C., Glanville, A. S., Millán, L., Deshler, T., Krotkov, N., and Toon, O. B.: Perturbations in stratospheric aerosol evolution due to the water-rich plume of the 2022 Hunga-Tonga eruption, Communications Earth & Environment, 3, 248,https://doi.org/10.1038/s43247-022-00580-w, 2022.

Zhu, Y., Akiyoshi, H., Aquila, V., Asher, E., Bednarz, E. M., Bekki, S., Brühl, C., Butler, A. H., Case, P., Chabrillat, S., Chiodo, G., Clyne, M., Falletti, L., Colarco, P. R., Fleming, E., Jörimann, A., Kovilakam, M., Koren, G., Kuchar, A., Lebas, N., Liang, Q., Liu, C.-C., Mann, G., Manyin, M., Marchand, M., Morgenstern, O., Newman, P., Oman, L. D., Østerstrøm, F. F., Peng, Y., Plummer, D., Quaglia, I., Randel,W., Rémy, S., Sekiya, T., Steenrod, S., Sukhodolov, T., Tilmes, S., Tsigaridis, K., Ueyama, R., Visioni, D.,Wang, X.,Watanabe, S., Yamashita, Y., Yu, P., Yu, W., Zhang, J., and Zhuo, Z.: Hunga Tonga–Hunga Ha′ apai Volcano Impact Model Observation Comparison (HTHH-MOC) project: experiment protocol and model descriptions. Geoscientific Model Development, 18(17), 5487–5512.

---

## Author Comment (AC1)

We thank the reviewers for their constructive and supportive comments, which have significantly improved the manuscript. We address all comments below.

Reviewer's comments are in black. Authors responses are in blue. Changes to the text are enclosed in double quotes.

**Reviewer 1:**

**Major comments**

Mc1. In Sect. 1 (Introduction), the authors summarize previous works that estimated the radiative forcing of the Hunga eruption. However, the distinction between the present study and these earlier studies is not explicitly articulated. Specifically, while the current and previous works (Sellitto et al., 2022; Zhu et al., 2022; Jenkins et al., 2023; Schoeberl et al., 2024; Stenchikov et al., 2025) all examine the radiative impacts of the Hunga eruption, it remains unclear in which aspects this study provides a novel contribution. For example, does the current study differ in methodology, model choice, experimental setup, dataset, study period, atmospheric level of the forcing, or spatial domain? A clearer discussion of the study's novelty would better engage the reader and also highlight the significance of the present work.

Our study is the first reporting multi-model analyses of the Hunga eruption following the HTHH protocols developed by the community in Zhu et al., 2025. We now highlight this better in the introduction.

"In this paper, we examine the radiative forcing resulting from these changes using a multi-model approach, by using several proposed HTHH-MOC experiments. Our goal is to highlight the sources of agreement and differences across models in estimating the magnitude of the forcing. Three different atmospheric setups are considered for the Hunga eruption simulation: (1) temperature and meteorology nudged to observations; (2) a free running meteorology setup with fixed sea surface temperature and sea ice extent; (3) and free-running meteorology setup with interactive ocean. The framework allows to distinguish the contributions from the direct interactions between the forcing agent and radiation, the rapid adjustments to these forcings, and slower climate feedback. Additionally, sensitivity studies performed within the nudged setup compare the effects of SO2-only and water vapor-only injections with the combined impact of both. The long-term evolution of Hunga water vapor and aerosols in the free-running models provides broader projections of the eruption climate impact. This comprehensive approach differentiates our study from previous work by integrating multiple experimental setups to better capture the complex interactions and feedbacks associated with the eruption."

MC2. The study has used GloSSAC for sAOD and SWOOSH for H2O to compare the model output with observation in Fig. 4. But the details about them are included very briefly in P.11 L.216–218. In addition, information about other observational datasets which are used for the simulations (if any) is missing. I recommend that authors create a dedicated data section which

outlines all the observational and meteorological datasets used in the study for the reproducibility of the results and to increase the reliability of the findings.

A new section, "Observational Datasets," has been added to describe the two datasets and explain our choice. Additional details are provided in the response to Major Comment #3.

"We use two observational datasets in this study: the Global Space-based Stratospheric Aerosol Climatology version 2.22 (GloSSAC, NASA/LARC/SD/ASDC) for zonal monthly-mean stratospheric aerosol optical depth (AOD), and the Stratospheric Water and Ozone Satellite Homogenized dataset version 2.6 (SWOOSH, Davis et al., 2016)  for water vapor.

GloSSAC provides a long-term and global record of stratospheric aerosol properties, including the stratospheric AOD at 525 nm used here. It is primarily based on Stratospheric Aerosol Gas Experiment (SAGE) measurements up to mid-2005, with Optical Spectrograph and Infrared Imaging System and Cloud-Aerosol Lidar and Infrared Pathfinder Satellite Observations used thereafter, and SAGE III extending the climatology to the present. Additional data from other satellites, as well as ground-based, airborne, and balloon-borne instruments, are included to fill observational gaps (Thomason et al., 2018; Kovilakam et al., 2020).

Although SAGE III/ISS data have limited spatial and temporal coverage, particularly during the first weeks following the Hunga eruption, they are considered robust because they rely on solar occultation and do not require assumptions about aerosol type or particle size distribution. In contrast, limb-scatter datasets such as Ozone Mapping and Profiler Suite Limb Profiler (OMPS-LP), used for comparison in Zhuo et al. (2025) depend on additional assumptions about aerosol properties (Kovilakam et al., 2025). For these reasons, GloSSAC is adopted as the primary observational reference, providing a consistent and reliable basis for evaluating modeled stratospheric aerosol evolution over longer timescales.

SWOOSH is a long-term and global record of stratospheric ozone and water vapor measurements from multiple satellite instruments (SAGE II, SAGE III, Halogen Occultation Experiment, Upper Atmosphere Research Satellite Microwave Limb Sounder, and Earth Observing System Aura Microwave Limb Sounder) spanning 1984 to the present."

MC3. One of the key arguments of this study is that the radiative forcing under the SO2 + H2O combined condition is higher than under the SO2 condition alone, and the authors have attributed this observation to enhanced sulfate aerosol growth in the presence of H2O. However, this claim is not sufficiently supported with evidence in the current manuscript. This point could be strengthened by including a plot of the aerosol size distribution under the two scenarios (SO2 + H2O and SO2 alone), which would provide direct support for the stated mechanism.

We agree that illustrating the early differences between the SO2andH2O and SO2-only simulations helps strengthen the discussion of the mechanisms leading to the larger radiative forcing in the combined scenario. We have expanded the supplementary material by adding a

figure showing the evolution of the vertical profile of the tropical effective radius under the different simulation scenarios (SO2andH2O, SO2-only, NoVolc). These additions support the discussion in the manuscript where we state that differences in IRF between the SO2andH2O and SO2-only are largest in the first few months, consistent with earlier work suggesting that the anomalously large water vapor injection favored faster early particle growth. Please see the response to minor comment #8-9 (MnC8 and 9) for the corresponding changes in the manuscript and the figures attached.

As noted in our response to Reviewer #2 (see MC1 and SC14), our manuscript doesn't provide full aerosol size distributions as it lies beyond the scope of this study, which focuses on the radiative response across models rather than a detailed microphysical evaluation. A more comprehensive modeling–observation comparison will be presented in a dedicated forthcoming paper.

MC4. The flow and discussions made in Sect. 3.1 and 3.2 are clear enough, but Sect. 3.3 is confusing and very difficult to follow. The primary reason for the confusion is the authors' abrupt shift to referencing multiple figures. This shift complicates the clarity of their argument. For example, the authors begin discussing Fig. 5 but abruptly shift to Fig. 6 without completing the explanation of Fig. 5. This pattern repeats throughout the section, creating a disorganized flow. Nearly seven figures are cited here, yet none of them are discussed in sufficient detail. I would recommend that authors reorganize this section by discussing each figure with relevant interpretations and scientific arguments, and move on to another figure. I would also suggest that the authors change the title of Sect. 3.3 to make it more descriptive.

We thank the referee for this comment and for pointing out issues related to the clarity and flow of Sect. 3.3.  We have substantially revised this section to improve readability and logical flow. We have also revised the order in which figures are discussed and moved some figures between Sect. 3.3 and the Supplementary Material. However, we consider that the current section title accurately reflects its scope and therefore we didn't change it.

"In this section we show a comparison of radiative forcing in the three different experiments, which only WACCM6-MAM conducted in full. This comparison provides useful insights to the different ways to define the radiative impacts of the plume between RF, ERF and IRF.

Figure 5 shows that, across all experiments and at the three atmospheric levels, the simulated clear-sky radiative forcing is negative and locally statistically significant in the Southern Hemisphere during the first two years following the eruption. The sign, as well as the spatial and temporal evolution, is consistent across the different model configurations and atmospheric levels considered. However, when including the atmospheric temperature adjustments in Fixed-SST and the ocean response in Coupled, the radiative forcing from aerosols and water vapor is substantially smaller than in the nudged simulation, particularly at TOA. This difference is more evident when averaging over the Southern Hemisphere (60°S to the equator), as shown in Fig. 6.

In panel 6a, the clear-sky IRF at TOA during the first 2 years after the eruption is -0.43 W m−2 in the nudged simulations, whereas in the free running experiments it is -0.27 W m−2 in both the coupled ocean and atmosphere-only cases. Although smaller than in the nudged simulations, these responses remain outside the range of natural variability, which is estimated in Fig. 6 as one standard deviation from the control ensemble (NoVolc). In general, while most of the results presented here are for the SH only, when considering global mean, the values are reduced by approximately half, reaching a value of -0.09 W m−2 at the TOA for the Coupled experiment in 2022-2023, which falls within the range of natural variability (see Fig. A7 and Table 4).

The reduced net radiative forcing relative to the nudged case arises from the inclusion of temperature and ozone adjustments. Indeed, differences in the simulated stratospheric AOD changes among the experiments are negligible (Fig. A8a–c), consistent with previous studies showing that atmospheric nudging does not necessarily improve the representation of the residual stratospheric circulation (Chrysanthou et al., 2019). Instead, the response is largely dominated by gas-driven variability, which is strongest in the Coupled experiment and includes a late-period negative radiative forcing in 2025 that is not present in the other simulations (Fig. 6a-b).

The separation of aerosol–radiation and gas–radiation contributions to the total radiative forcing (Fig. 7) shows that the negative TOA forcing during the first two years following the eruption is primarily driven by the aerosol contribution from Hunga, partially offset by a positive contribution from changes in gases. This behavior contrasts with the nudged simulations, where the gas–radiation interaction produces a negative forcing (Fig. 3). In the free-running simulations, the gas contribution is strongly influenced by natural variability (red line in Fig. 7), particularly for all-sky radiative forcing, which limits the detectability of the forced response without large ensemble sizes (Fig. A9).

In the second half of 2025, as the tropical stratospheric AOD returns to background levels (Fig. A8a–c), only the Coupled experiment exhibits a significant negative radiative forcing in the tropics at both TOA and TROP, with magnitudes comparable to those observed during the first post-eruption year (Fig. 6a and b), while the forcing at SURF becomes positive. This late-period forcing in the Coupled experiment is primarily driven by gas–radiation interactions, with significant negative clear-sky values emerging in 2025 (Fig. 7c).

The gas radiative forcing results not only from Hunga-induced changes in stratospheric water vapor (which are similar across the three experiments; Fig. A8d–f) but also from changes in stratospheric temperature and ozone, as well as associated dynamical adjustments that modulate both, particularly in the lower stratosphere. These contributions are further explored in Fig. 8. The gas–radiation interaction is dominated by the LW component (Fig. 8a–c), with differences among the atmospheric configurations driven primarily by temperature and dynamical responses: Zhuo et al. (2025) has shown an upper-level cooling due to the water vapor and a lower stratospheric level warming from LW absorption from the sulfate aerosols in the free-running simulations.

During the first two years following the eruption, the TOA gas radiative forcing is positive in the Coupled simulation, more variable in Fixed-SST, and negative in the Nudged case (Fig. 8a–c). This behavior reflects the combined effects of increased water vapor in the middle stratosphere and reduced ozone in the lower stratosphere. When temperature and dynamical changes are excluded, as in the Nudged simulation, the water vapor increase and ozone decrease (black curves in Fig. 8f and i) enhance  outgoing LW radiation at TOA, resulting in a negative forcing. In contrast, when temperature adjustments are included, as in the Coupled and Fixed-SST simulations, the cooling at higher altitudes where water vapor anomalies peak (black curves for water vapor and red curves for temperature in Fig. 8d and e) reduces outgoing LW radiation, yielding a positive TOA forcing during the first two post-eruption years. Furthermore, the Coupled shows a distinct evolution in the tropical lower stratosphere, characterized by increased ozone and warming during 2022–2023, followed by decreased ozone and cooling in 2025 (panel 8g). Because ozone in the lower stratosphere acts as a greenhouse gas, these changes contribute to an additional positive radiative forcing during the first two years and a negative forcing in 2025, when water vapor anomalies have largely returned to background levels. The close correlation of changes in ozone and temperatures in this region, not seen in Fixed-SST (panel 8h), is strongly indicative of their dynamical origin, suggesting an associated decrease in tropical upwelling in 2022-2023 and increase in 2025.

As discussed in (Bednarz et al., 2025), the coupled ocean WACCM6-MAM simulations shows a significant modulation of the El Niño–Southern Oscillation (ENSO) variability by the eruption, with La-Nina like response in 2022-2023 and an El-Nino like response in 2025. In general, ENSO is an important driver of interannual variability in tropical upwelling, which in turn modulates lower-stratospheric temperatures and ozone (Randel et al., 2009), and can therefore exert a substantial influence on the overall radiative forcing. "

MC5. Another major issue is that most of the figures are underdiscussed, i.e., in any figure, only two to four panels are discussed, leaving out the rest of the panels untouched, which questions their very purpose. Initially, I thought this specific issue was minor and restricted to Fig. 2 alone (please see minor comment no. 7 for more details). But later, I realized it was prevalent among most of the figures, making it a major issue. I highly recommend that the authors increase discussion on each figure (if not all panels, try to explain the most panels), and connect them to the objective of the study. If some figures are not worth discussing or if the authors feel that such discussion may impact the manuscript flow, they can be safely moved to the supplementary section.

We have expanded the discussion of each figure and addressed the previously underexplained panels. In addition, we have ensured that all relevant panels and supplementary figures are properly referenced in the text. These revisions are detailed in our responses to the minor comments.

**Minor comments**

MnC1. I have a suggestion regarding the title of the manuscript, "Multi-model analysis of the radiative impacts of the 2022 Hunga eruption indicates a significant cooling contribution from the volcanic plume." I appreciate that the authors have highlighted the key finding (i.e., the cooling effect of the Hunga eruption) in the title. However, since previous studies have already established that the Hunga plume resulted in a net cooling, the current title does not appear very compelling and intriguing. I believe it would be more impactful to emphasize the amplification of this cooling (net radiative forcing) due to the growth of sulfate aerosols caused by the water-enriched stratosphere. Also, the title looks lengthy at present. So, if the authors are changing the title, it is better to be concise.

We changed the title to "Multi-model analysis of the impact of water vapor on the radiative forcing of volcanic aerosols after the 2022 Hunga Eruption"

MnC2. P.2 L.36: The authors have used the phrase "net warming" to refer to positive forcing. And at L.38 and L.40, the phrase "negative forcing" is used to refer to net cooling. These lines (L.35 to L.40) highlight the contrasting findings among the previous studies, as any good introduction should. But inconsistent usage of the phrase referring to the radiative forcing decreases the readability of these sentences. Hence, I would like to suggest that the authors use either "net warming" and "net cooling" or "positive forcing" and "negative forcing" to increase readability.

Because Jenkins et al. (2023) focuses primarily on surface temperature responses, we have chosen to keep the term "net warming" in this specific context. Elsewhere in the manuscript, where we are referring explicitly to radiative forcing rather than surface temperature response, we consistently use the terms "positive" and "negative" forcing without implying an associated surface temperature impact.

MnC3. I appreciate the authors for providing details about the types of experiments and models employed in this study in Sect. 2.1 in P.3 and about the injection parameters through Table 1. However, for a common reader like me, it takes two to three reads to get clarity and understand the method itself. It also disrupts the flow of the manuscript. To rectify it, I highly encourage the authors to add a graphical representation of the methods or a table which clearly describes all the experiments and models concisely. In addition, even though the current manuscript is part of the HTHH-MOC project and details of the employed methods are already furnished in the main paper (Zhu et al., 2025), I believe it is important to furnish minimal information (if not extensive) about the experiment design in the current manuscript for enhanced clarity.

The section has been revised to include a sentence clarifying the different injection-setting simulations (SO2andH2O, SO2only, H2Oonly) included in each experiment (Nudged, Fixed-SST, and Coupled) and a new table has been included (see figure below).

"Each experiment includes at least one simulation that involves the combined injection of 0.5 Tg of SO2 and 150 Tg of H2O ("SO2andH2O"), alongside a control simulation without any injections ("NoVolc"). The Nudged experiment also includes single-forcing simulations that inject only SO2 ("SO2only") or only H2O ("H2Oonly"). The location, altitude, and amount injected are different between models in order to better match the observed plume in the first couple of days, but they remain consistent across experiments between models. Models account for the interactive coupling between aerosol, water vapor, radiation, and dynamics, allowing the fast descent due to water vapor longwave cooling (Sellitto et al., 2022) to be simulated through this coupling. A summary of the experiments is provided in Table 1, and detailed model-specific injection settings are listed in Table 2."

**Table 1.** Summary of experiments

| Experiment name (former name in Zhu et al., 2025) | Meteorological configuration | Year simulated | Subset of experiment included |
|---|---|---|---|
| Nudged (Exp2a) | Nudged wind and temperature, fixed sea surface temperatures | 2022-2023 | NoVolc SO2andH2O SO2only H2Oonly |
| Fixed-SST (Exp1_fixedSST) | Free running meteorology, fixed sea surface temperatures | 2022-2031 | NoVolc SO2andH2O |
| Coupled (Exp1_coupled) | Free running meteorology, atmospheric-ocean coupling | 2022-2031 | NoVolc SO2andH2O |

**Table 2.** Injection parameters. Adapted from Table 7 in Zhu et al. (2025)

| Model | $H_2O$ injected | $H_2O$ altitude | $SO_2$ injected | $SO_2$ altitude | Injection location |
|---|---|---|---|---|---|
| WACCM6-MAM | 150 Tg | 25-35 km | 0.5 Tg | 20-28 km | 22-14°S, 182-186°E |
| WACCM6-CARMA | 150 Tg | 25-35 km | 0.5 Tg | 26.5-36 km | 22-6°S, 182.5-202.5°E |
| MIROC-CHASER | 150 Tg | 25-30 km | 0.5 Tg | 25-30 km | 22-14°S, 182-186°E |
| CAM5-CARMA | 150 Tg | 25-35 km | 0.5 Tg | 20-28 km | 22-14°S, 182-186°E |
| UKESM | 150 Tg | 25-30 km | 0.5 Tg | 25-30 km | 22-14°S, 182-186°E |

MnC4. In P.5 L.109–124, a few radiative forcing (RF) formulas are mentioned. Firstly, please consider adding equation numbers for all of them. Secondly, as the authors have clearly mentioned that "Unless otherwise specified, all values are assumed to be calculated under Clear-Sky conditions" at L.107, it is not essential to include 'clear' as a superscript in each term. In addition, the definition of '$F_{volc\ clear}$' and '$F_{control}$

*clear'* varies in each formula, and their description is given at the end. However, it looks a bit odd to write them in this fashion. I suggest the authors consider rewriting the equation with more detailed subscripts and superscripts, something like what is given below.

We have partially adopted the reviewer's recommendation regarding the revision of the RF definitions to be consistent with the experiment definitions in table 1. The revised section is now as follows:

"There are important differences among the radiative forcing estimates of volcanic eruptions derived from nudged simulations, which provide the instantaneous radiative forcing (IRF; Eq. 1); free-running atmosphere-only simulations, which yield the effective radiative forcing (ERF; Eq. 2), following the definition of Forster et al. (2016); and fully coupled simulations with an interactive ocean, which are generally referred to as radiative forcing (RF; Eq. 3).
[...]
As a result, the IRF from aerosols and water vapor, either combined or individually, can be approximated using the corresponding Nudged experiments (Eq. (4) and (5), respectively). In the case of WACCM6-MAM simulations, both methodologies have been applied. Notably, in contrast to previous studies, all forcing estimates presented here treat stratospheric water vapor as a forcing rather than a feedback, due to its direct injection.
In summary, the following sections explain how the different radiative forcing estimates are calculated. Radiative forcing is calculated under both Clear-Sky (CS, without clouds) and All-Sky (AS, including the effects of clouds) conditions. Unless otherwise specified, all values are assumed to be calculated under Clear-Sky conditions.

$$IRF = F_{SO2andH2O,Nudged} - F_{NoVolc,Nudged} \tag{1}$$

$$ERF = F_{SO2andH2O,Fixed-SST} - F_{NoVolc,Fixed-SST} \tag{2}$$

$$RF = F_{SO2andH2O,Coupled} - F_{NoVolc,Coupled} \tag{3}$$

$$IRF_{aerosol} = F_{SO2only,Nudged} - F_{NoVolc,Nudged} \tag{4}$$

$$IRF_{gas} = F_{H2Oonly,Nudged} - F_{NoVolc,Nudged} \tag{5}$$

In WACCM6-MAM, a second estimate of instantaneous radiative forcing for aerosols and gases is derived through a double radiation call (Eq. (6) and (7), respectively). Within this approach, $F^{clean}$ denotes the Clean-Sky calculation, which excludes aerosols.

$$IRF_{aerosol} = (F_{SO2andH2O,Nudged} - F_{NoVolc,Nudged}) - (F^{clean}_{SO2andH2O,Nudged} - F^{clean}_{NoVolc,Nudged}) \tag{6}$$

$$IRF_{gas} = F^{clean}_{SO2andH2O,Nudged} - F^{clean}_{NoVolc,Nudged} \tag{7}$$

"

MnC5. In Fig. 1 (in all panels, except H2O_only cases), negative RF (~-0.4 Wm-2) is observed at the high-latitude (60°–90°S) from November 2022 to February 2023. But thereafter, the RF becomes close to 0 Wm-2 during April–September 2023 (Austral winter). This feature is

consistent across all models and experiment setups and also appears in the Austral winter of 2022. Since the authors have clearly stated in P.6 at L.148 that IRF also considers the polar stratospheric clouds (PSCs), these RF values correspond to PSC as well as other aerosols too (as it is a clear-sky condition and not clean-sky). In this scenario, is RF ~0 Wm-2 caused by the formation of PSC during the Austral winter? Is it because net cooling by stratospheric aerosols is countered by net warming by the PSC? Please describe.

The RF equals zero during the Austral winter because the strong reduction in solar insolation makes the shortwave RF very small, comparable to the small and positive longwave RF arising from changes in aerosol thermal emissions. This longwave contribution coincides with the peak in stratospheric AOD over Antarctica, while the shortwave component is also influenced by seasonal solar insolation. See response to comment MnC7 for more details.

MnC6. Please explain the cause of the data gap in Fig. 1e and m.

The data gap in Fig. 1e and 1m occurs because the model did not produce output for those months, either because the simulation did not run through the full period or because the relevant variables were not written to the output. We have updated the caption of Fig. 1 to include the sentence: "The data gap is due to missing model output for those months."

MnC7. The authors have estimated the RF at three key atmospheric levels: TOA, TROP, and SURF, and done an excellent job of explaining physical implications and reasons for choosing these three levels in P5 at L.126–132. These results are shown in Fig. 2 as well, with interpretation included on P.6 at L.156–162. While the discussions of RF at TOA and TROP are briefly touched on, the SURF RF has not been addressed. If the SURF RF is not required for the discussion, then its presence becomes questionable. Moving less important plots (panels) to the appendix or supplementary section may help the readers to be focused on the main argument which authors are trying to make. In addition, the discussion could be strengthened by elaborating on the causes of the observed variation in RF across latitude bands and their temporal evolution. (i) To be specific, for the case of SO2andH2O, the global mean (Fig. 2a, black
solid line), and 60°S–Eq mean RF at TOA (Fig. 2d) show that the strongest negative RF occurred almost at the end of 2022. In contrast, at the TROP and SURF, the strongest negative RF occurred much earlier, around February 2022. What could be the possible physical mechanism of this result? (ii) In the high-latitude band (60–90°S), RF is close to 0 Wm-2 during the first six months of 2022 at all three levels (TOA, TROP, and SURF; third row in Fig. 2). However, a sudden strong negative RF exceeding -0.5 W m-2 appears in January 2023, which rapidly decays and returns to ~0 W m-2 by the peak Austral winter of 2023. Following this, another negative RF peak occurs at the end of 2023. What physical processes drive this temporal variation, particularly the ~0 W m-2 RF during the Austral winter? This point is directly connected to my earlier comment on Fig. 1 (see major comment no. 5). A solid discussion on these aspects could potentially strengthen the discussion and make it Comprehensive.

A more detailed discussion of the three levels has been added in the first part of the section, with references to the corresponding figure in the appendix, as follows:

"The IRF includes all optically active components, such as stratospheric aerosols, water vapor, ozone, and polar stratospheric clouds (PSCs). For model intercomparison, we use clear-sky forcing to minimize uncertainties associated with aerosol–cloud interactions that arise from differences in cloud parameterizations among models.

All five models show qualitative agreement in the spatial distribution of the IRF under clear-sky conditions at TOA (Fig. 1), at TROP (Fig. A1), and at SURF (Fig. A2). At each level, the response is primarily located in the Southern Hemisphere (SH); therefore, most of the following analyses will focus on the SH only.

A consistent pattern of negative forcing appears across models in both the SO2andH2O and SO2only experiments (first and second columns in Figs. 1, A1, A2). This forcing peaks in the tropics during the first few months following the eruption and then moves to mid- to high-southern latitudes by 2023, with substantial negative forcing persisting through the end of 2023 in the SH. Likewise, all models simulate a negligible IRF from the H2Oonly experiment (third column of the same figures). Therefore, the models consistently attribute nearly all of the radiative forcing to the aerosol perturbations rather than to the injected water vapor.

Global and hemispheric means in Fig. 2 show the multi-model average and inter-model spread, and highlight the magnitude of the IRF in each injection experiment. These means further clarify that the IRF from H2Oonly is negligible: it is slightly negative at TOA (first column) and slightly positive at TROP and SURF (second and third columns). This behavior results from the vertical distribution of water vapor in the stratosphere and the lack of stratospheric temperature adjustment, which together increase outgoing longwave radiation while enhancing downward longwave re-emission. Especially at the surface, the IRF remains positive but very small, as the radiative forcing is dominated by sensible and latent heat fluxes, which are controlled by the prescribed surface temperature in these simulations.

For the aerosol forcing, which thus constitutes essentially the entirety of the Hunga radiative forcing, the models generally agree on its temporal evolution but exhibit substantial differences in its magnitude. The multi-model mean TOA IRF from SO2andH2O is -0.35 ± 0.12 W/m2 over 60∘S-0∘ (2022–2023 average), where the inter-model spread is roughly one third of the mean magnitude. The global mean forcing is approximately half of the 60∘S-0∘ values,reflecting the strong SH dominance of the perturbation. Over the high southern latitudes (60∘-90∘S), the mean TOA IRF is -0.26 ± 0.23 W/m2, indicating large uncertainty, with the standard deviation nearly matching the mean (September 2022–December 2023 average). Comparing the injection experiments, the IRF from SO2andH2O is generally comparable to SO2only at TOA, but less negative at TROP and SURF: over the 60∘S-0∘, TROP IRF is -0.32 ± 0.13 for SO2andH2O versus -0.43 ± 0.12 W/m2 for SO2only; at the surface, these values are -0.29 ± 0.12 versus -0.36 ± 0.11 W/m2.

The shortwave (SW) component of the IRF follows the evolution of stratospheric aerosol optical depth (AOD, Fig. A3) but is additionally modulated by the strong seasonality of solar insolation, producing relative minima during the Austral winter. This is particularly evident at high southern latitudes (Fig. 2g–i), where the small negative SW IRF is partially offset by a positive longwave

(LW) contribution. Generally, the LW signal coincides with the peak in stratospheric AOD, which, over the high southern latitudes, occurs during the Austral winter of 2023.
The SW IRF is similar between TOA and TROP but smaller in magnitude at SURF in both SO2andH2O and SO2only experiments. However, at all levels, the SW IRF is more negative in SO2andH2O during the first few months after the eruption and then becomes less negative compared to SO2only, consistent with the evolution of stratospheric AOD in the two experiments (panels j and k in Fig. A3). The differences in the net IRF between the two experiments decrease at TOA but increase at TROP
due to the different LW responses to water vapor, which is slightly negative at TOA and positive at TROP."

MnC8. The authors have written, "The fact that the forcing becomes similar later on and shows a slight reduction in the last few months of 2023, may suggest a balance between larger particles and shorter lifetime for the two cases" in P.7 at L.170–172. Could it also be due to the polar transport of volcanic emissions from mid-to-high latitude, as evident from Fig. 1?

We added a supporting figure in the Supplementary Information showing the evolution of the AOD change. It demonstrates that during the first few months the AOD increases more rapidly in the SO2andH2O case than in the SO2only case (panels j-i, solid and dashed lines, respectively; the black line shows the multi-model mean). After this initial period, however, the AOD in the SO2andH2O becomes smaller than in the SO2only, regardless of the averaging region. This behavior indicates that the later similarity and slight decline in forcing are not driven by polar transport but by the differing microphysical evolution in the two scenarios.

[Figure]

Figure A3. Time series of zonal mean stratospheric aerosol optical depth anomalies (perturbed minus control) from five models: WACCM6-MAM (a, f), WACCM6-CARMA (b, g), MIROC-CHASER (c, h), CAM5-CARMA (d, i), and UKESM (e). The first two rows correspond to perturbation scenarios from SO2andH2O, and the second row shows SO2only. The third row includes regional averages of the same quantity for two latitude bands: 90°S-N (j), 60°S–equator (k) and 60–90°S (l).

MnC9. P.7 L.185–187: This is one of the interesting and important findings made in the present study and also included in the abstract. However, without a plot or supporting data, it appears speculative. To strengthen this claim, consider adding a bar plot (or any other suitable plot authors wish) comparing aerosol sizes under SO2andH2O and SO2 experiments, for both clear-sky and all-sky scenarios. This would provide clearer evidence to support the finding.

Also in response to Referee #2's comment, we have included the discussion in the first bullet point, relating it to the previously mentioned figure on stratospheric AOD (Fig. A3) as well as a new figure (Fig. A5). This figure shows the time series of the tropical effective radius, where available, for the SO2andH2O, SO2only, and NoVolc, and it highlights the faster initial particle growth and more rapid return to background values in SO2andH2O compared to SO2only.

"The IRF from the SO2andH2O and SO2only simulations (black and orange lines in panel 3a) show similar behavior from late 2022 onward. However, substantial differences are evident during the first six months (on the order of 20% in WACCM6-MAM, panel 3c), with the exact duration being model dependent (Fig. A4). This is in agreement with previous studies indicating that the co-injection of stratospheric water vapor promotes faster particle growth to optically efficient scattering sizes during the initial months (panels a–c for SO2andH2O versus panels d-f for SO2only in Fig. A5). As a result, SO2andH2O produces a larger initial stratospheric AOD (Fig. A3 j-k) and corresponding negative radiative forcing. Later on, the forcing becomes similar due to a compensating effect: SO2only shows a larger stratospheric AOD, while SO2andH2O shows a stronger gas contribution (negative at TOA). The SO2andH2O forcing becomes slightly weaker only in the last few months of 2023, when the gas IRF in the two simulations becomes comparable (panel 3c)."

[Figure]

Figure A5. Time series of effective radius averaged over 30°S and the equator for (a, d, g) WACCM6-MAM, (b, e, h) WACCM6-CARMA, and (c, f, i) MIROC-CHASER. Each row corresponds to a different scenario from the nudged experiment: the first column shows SO2andH2O, the second column SO2only, and the third column NoVolc.

MnC10. P.7 L.194–195: Any possible explanation for the discrepancies between the models at the tropopause?
We expanded the explanation after discussing the changes in stratospheric AOD, water vapor and ozone. However the answer is speculative since for MIROC-CHASER we can't evaluate the forcing agent contributions separately. We added the following section:

"As with the instantaneous radiative forcing, the gas response varies depending on the atmospheric level at which it is calculated and the vertical distribution of the gases. Additionally, temperature adjustments are included here. We speculate that the stronger stratospheric AOD anomaly in WACCM6-MAM, along with the stronger water vapor and ozone anomalies in MIROC-CHASER, may either offset the radiative forcing at the top of the atmosphere (TOA) or amplify it at the tropopause (TROP). This could explain the similar forcing observed in Fig. 4a

and the larger differences seen in Fig.4b. This aspect is only further explored in Sect. 3.3 for WACCM6-MAM, where the availability of separate aerosol and gas radiative contributions allows for a clearer disentangling of each factor."

MnC11. P.10 L205: What does 'cloud' refer to in this context? Are the authors referring to the aerosol plume from the Hunga eruption? If so, please use the term 'aerosol plume' for consistency, since this phrase is already used at l.204. If not, kindly clarify what is meant by 'cloud' here.
We meant "volcanic cloud," which is now explicitly stated at line 205 to improve clarity.

MnC12. P.13 L.249: The Fig. A8 is cited before Figs. A1–A7 are not even mentioned. Except Fig. A7, which is cited at L.275 in P.14, Figs. A1–A6 are never cited at all. If these figures are not even needed to be cited in the main text and do not contribute to the discussion, their inclusion even in the appendix is questionable.
We revised the manuscript to ensure that all figures are cited in the correct order, as mentioned in response to the major comment #5.

**Technical comments:**

TC1. Throughout the manuscript, the unit for the radiative forcing is written as 'W/m2.' I would like to suggest changing them to 'Wm-2' both in the main text and plots as per the ACP template.
Done.

TC2. P.4 L91–92: The instantaneous radiative forcing is already abbreviated as IRF at L.82. Hence, it is not necessary to abbreviate it again. Please correct.
Done.

TC3. P.5 P.120: Write 'instantaneous radiative forcing' as IRF.
Done.

TC4. P.6 L.134: Please write 'Section 3.1' as 'Sect. 3.1'
Done.

TC5. P.6 L.138: The first paragraph of Sect. 3 provides the outline of the result and discussion section. While doing so, it would be better if the corresponding subsections are cited at the appropriate place, as shown below.
Done.

TC6. "Following that, we provide analyses of the two models which provided the 10-year long free-running simulations with prescribed climatological SSTs and sea ice (Sect. 3.2): these analyses allow us to understand the long-term behavior of the forcing as well as include the combined chemical and dynamical impacts and temperature Adjustments."

Done.

TC7. "Finally, we complete those with an analysis of fully-coupled simulations in WACCM6-MAM (Sect. 3.3)."
Done.

TC8. P.6 L147: Replace 'timeseries' with 'time series'
Done.

TC9. P.7 L.165: Replace 'Section 2.2' with 'Sect. 2.2'
Done.

TC10. P.7 L189: Replace 'Fig.4' with 'Figure 4'
Done.

TC11. P.7 L189: Does the 't' in the formula refer to 'time'? Please clarify.
We modified the equation as a response to referee #2 and we clarified it as follows

"[..] the moving average forcing, calculated as $\frac{1}{N_t}\sum_{i=1}^{N_t} RF_i$, where $N_t$ represents the number of months elapsed, and $RF_i$ is the radiative forcing at month i. "

TC12. In the caption of Fig. 2, it is mentioned that the first row plots correspond to 60°S–90°N. But in the title of the panel a–c, it is written as 90°N–S. Please correct.
Done.

TC13. P.7 L200: The stratospheric aerosol optical depth is abbreviated as 'sAOD' at L.199 and referred again as 'stratospheric AOD' at L.200. Once an abbreviation is introduced, it should be used consistently throughout the main text (except in figure captions and the conclusion). Consistency in terminology will improve both readability and the overall flow of the manuscript.
We decided to use the term 'stratospheric AOD' throughout the text, but we kept sAOD in the figure due to space constraints in figure titles.

TC14. In Fig. 3d and e, please make a common y-axis limit. At present, the panel d has a slightly wider ylimit than panel e.
Done.

TC15. In the caption of Fig. 3, replace 'Section 2.2' with 'Sect. 2.2'
Done.

TC16. P.11 L.222: Replace "(Zhuo et al., 2025)" with "Zhuo et al., (2025)"
Done.

TC17. P. 12: In Fig. 4's caption, clearly indicate what the green and magenta shading region (panels a to c) represent.

Done: "Solid lines represent the ensemble mean, and shaded areas represent ±1 standard deviation across ensemble members. Each shaded area uses the same color as the line representing that model, as shown in the legend."

TC18. P.11 L.232: Replace 'Fig. 5' with 'Figure 5'
Done.

TC19. P.11 L.234–235: Replace 'stratospheric aerosol optical depth' with 'sAOD' as the acronym is already introduced.
Done as in TC13.

TC20. In the caption of Fig. 4, replace 'timeseries' with 'time series' and in the first line write 'effective radiative forcing' as 'effective radiative forcing (ERF)'.
Done.

TC21. In Fig. 4 (e, f, h, i, and k), 'Altitude (hPa)' is given as the ylabel. Usually, altitudes are given in units of m or km. Is it supposed to be 'Pressure (hPa)'? Please check and correct.
Done.

TC22. P.13 L.262: What does LW mean here? Please expand it.
LW (longwave) is now defined in section 3.1 (see answer to MnC7)

TC23. P.13 L270: Expand ENSO.
Done.

TC24. Figures 6, 7, and 8, and Table 3 appear in the Conclusion Sect. 4. I suggest the authors move them to Sect. 3.3, where relevant discussion using these figures and tables is made.
Done.

References

Jenkins, S., Smith, C., Allen, M., and Grainger, R.: Tonga eruption increases chance of temporary surface temperature anomaly above 1.5 °C, Nature Climate Change, 13, 127–129, https://doi.org/10.1038/s41558-022-01568-2, 2023.

Sellitto, P., Podglajen, A., Belhadji, R., Boichu, M., Carboni, E., Cuesta, J., Duchamp, C., Kloss, C., Siddans, R., BÚgue, N., Blarel, L., Jegou, F., Khaykin, S., Renard, J. B., and Legras, B.: The unexpected radiative impact of the Hunga Tonga eruption of 15th January 2022, Communications Earth & Environment, 3, 288, https://doi.org/10.1038/s43247-022-00618-z, 2022.
Schoeberl, M. R., Wang, Y., Taha, G., Zawada, D. J., Ueyama, R., and Dessler, A.: Evolution of the Climate Forcing During the Two Years After the Hunga Tonga-Hunga Ha'apai Eruption, Journal of Geophysical Research: Atmospheres, 129, e2024JD041 296, https://doi.org/https://doi.org/10.1029/2024JD041296, e2024JD041296

2024JD041296, 2024.

Stenchikov, G., Ukhov, A., and Osipov, S.: Modeling the Radiative Forcing and Atmospheric Temperature Perturbations Caused by the 2022 Hunga Volcano Explosion, Journal of Geophysical Research: Atmospheres, 130, e2024JD041 940, https://doi.org/https://doi.org/10.1029/2024JD041940, e2024JD041940 2024JD041940, 2025.

Zhu, Y., Bardeen, C. G., Tilmes, S., Mills, M. J., Wang, X., Harvey, V. L., Taha, G., Kinnison, D., Portmann, R. W., Yu, P., Rosenlof, K. H., Avery, M., Kloss, C., Li, C., Glanville, A. S., Millán, L., Deshler, T., Krotkov, N., and Toon, O. B.: Perturbations in stratospheric aerosol evolution due to the water-rich plume of the 2022 Hunga-Tonga eruption, Communications Earth & Environment, 3, 248,https://doi.org/10.1038/s43247-022-00580-w, 2022.

Zhu, Y., Akiyoshi, H., Aquila, V., Asher, E., Bednarz, E. M., Bekki, S., Brühl, C., Butler, A.H., Case, P., Chabrillat, S., Chiodo, G., Clyne, M., Falletti, L., Colarco, P. R., Fleming, E., Jörimann, A., Kovilakam, M., Koren, G., Kuchar, A., Lebas, N., Liang, Q., Liu, C.-C., Mann, G., Manyin, M., Marchand, M., Morgenstern, O., Newman, P., Oman, L. D., Østerstrøm, F. F., Peng, Y., Plummer, D., Quaglia, I., Randel,W., Rémy, S., Sekiya, T., Steenrod, S., Sukhodolov, T., Tilmes, S., Tsigaridis, K., Ueyama, R., Visioni, D.,Wang, X.,Watanabe, S., Yamashita, Y., Yu, P., Yu, W., Zhang, J., and Zhuo, Z.: Hunga Tonga–Hunga Ha′apai Volcano Impact Model Observation Comparison (HTHH-MOC) project: experiment protocol and model descriptions. Geoscientific Model Development, 18(17), 5487–5512, 2025.

**Reviewer 2:**

**Major comments**

MC1) The large radiative forcing efficiency of the relatively modest SO2 emissions of the Hunga eruption are attributed to specific aerosol size distributions of its plume, in this manuscript. By the way, the aerosol size distribution outputs of the model runs are not shown or discussed in the manuscript. There are also observations available for these size distributions, that can be used as a comparison of what obtained here with models. I strongly suggest introducing such discussions/comparisons in the next manuscript version. Please also see SC14.

In response to this comment and SC14, we have expanded the supplementary material by adding a figure showing the evolution of the vertical profile of the tropical effective radius under the different simulation scenarios (SO2andH2O, SO2-only, NoVolc). These additions support the discussion in the manuscript where we state that differences in IRF between the SO2andH2O and SO2-only are largest in the first few months (model dependent), consistent with earlier work

suggesting that the anomalously large water vapor injection favored faster early particle growth. Please see the response to SC15 for the corresponding changes in the manuscript and the figures attached.

Regarding the reviewer's request to include full modeled aerosol size distributions and direct comparisons with observational constraints, as discussed in response to SC14, we emphasize that such an in-depth microphysical evaluation lies outside the primary scope of this study. The focus of the present paper is on the radiative response to the Hunga eruption across multiple climate models and meteorological configurations, rather than a comprehensive assessment of microphysical processes or an observational intercomparison. These aspects are being addressed as part of the larger coordinated Hunga modeling effort, and will be presented in a dedicated, forthcoming paper that specifically targets the microphysical evolution and observational comparisons.

We believe the newly added supplementary figures strengthen the manuscript by providing the key size-evolution information needed for interpreting the radiative forcing results, while keeping the scope aligned with the study's objectives.

MC2) Also linked to MC1, I don't agree with the general interpretation given here for the unusually large radiative forcing efficiency of the Hunga aerosol plume. In different parts of the text, it is mentioned that the large radiative forcing efficiency is due to larger particle sizes than what generally is found in eruptive plumes, which is quite the opposite of what shown by Li et al. (2024) and Sellitto et al. (2025) (see more details in SC10 and 14). The Authors mention the work of Li et al. (2024) but this appears very late in the text ad contradicts what is said in the Introduction, while the work of Sellitto et al. (2025) is not even mentioned. All this interpretation must be clarified in the revised text.

As clarified in our response to SC14 and SC27, we have tried to clarify that our manuscript does not claim that the Hunga eruption produced aerosols larger than those from eruptions such as Pinatubo (points raised in SC10). Rather, our discussion now more clearly states that the large quantities of water vapor injected by Hunga favored faster particle growth in the initial months, meaning growth relative to background conditions, sufficient to reach sizes that are optically efficient at scattering radiation, which implies that promote the formation of particles in the accumulation mode rather than the coarse mode.

When referring to Zhu et al. (2022), the phrase 'increased size of aerosols' does not imply the formation of coarse particles or particles larger than those produced by major explosive eruptions. We also do not see any contradiction with the statements made in the Conclusions, as we are referring specifically to particle scattering efficiency and the resulting stratospheric AOD, without making claims about aerosol sizes. In this context, we now cite Li et al. (2024), who attribute the unusually high radiative forcing efficiency primarily to the higher injection altitude and the enhanced co-emission of water vapor.

MC3) I disagree on the use of GloSSAC-only observational data to analyse the capabilities of the model runs to describe the peak values of the sAOD. The GloSSAC is strongly based on SAGE III/ISS which, due to its scarce spatiotemporal coverage, especially at the very initial dispersion phases after the event, might underestimate this peak. Other spatiotemporally denser datasets, like the one from OMPS-LP, should be used, in addition to GloSSAC. See also SC22 and 23.

We acknowledge that the SAGE III/ISS data are sparse in space and time, particularly during the first weeks after the eruption. This limitation indeed affects the ability of GloSSAC to capture the high peak in sAOD immediately following the event. We have clarified this point in the revised manuscript in response to the Specific Comment #23 (SC23),  to avoid any ambiguity regarding the origin of the missing initial peak, and we add more details from the analysis of stratospheric AOD evolution by Zhuo et al. (2025; https://doi.org/10.5194/acp-25-13161-2025), where both dataset are discussed.

However, we choose to keep GloSSAC as the main observational reference for the model evaluation. SAGE III/ISS aerosol extinction measurements are based on solar occultation and do not require assumptions about aerosol type or particle size distribution for individual retrievals. By contrast, limb-scatter datasets such as OMPS-LP rely on additional assumptions related to aerosol microphysics and scattering properties (Kovilakam et al., 2025; https://doi.org/10.5194/acp-25-535-2025). The different OMPS-LP algorithm gives substantially different sAODs especially at the start (see Ch. 3 of the assessment, doi: 10.34734/FZJ-2025-05237). As a result, SAGE III/ISS–based products are generally considered more reliable for quantitative sAOD comparisons, especially when focusing on absolute values rather than short-term variability. The goal of this study is not to fully describe the very early dispersion phase of the Hunga aerosol cloud, but to assess the modeled stratospheric aerosol evolution using a climatological dataset. Including OMPS-LP would add extra uncertainty related to retrieval assumptions that are beyond the scope of this work, and differences between the two datasets are already assessed in Zhuo et al. (2025).
The reasons for our observational choice has been included in a new section "Observational dataset":

"GloSSAC is a long-term and global record of stratospheric aerosol properties,  including the stratospheric aerosol optical depth at 525 nm used in this study. The dataset is primarily based on SAGE measurements up to mid-2005, with OSIRIS and CALIPSO observations used thereafter, and SAGE III/ISS extending the climatology to the present. Additional data from other satellites, as well as ground-based, airborne, and balloon-borne instruments, are included to fill observational gaps (Thomason et a., 2018; Kovilakam et al., 2020).
Although SAGE III/ISS data have limited spatial and temporal coverage, particularly in the first weeks after the Hunga eruption, they are considered robust because they rely on solar occultation and do not require assumptions about aerosol type or particle size distribution. In contrast, limb-scatter datasets such as OMPS-LP, used for comparison in (Zhuo et al.,2025), rely on additional assumptions about aerosol properties (Kovilakam et al., 2025). For these reasons, GloSSAC is adopted here as the primary observational reference, providing a

consistent and robust basis for evaluating the modeled evolution of stratospheric aerosols over longer timescales."

MC4) If the Authors support the possibility that a larger amount than the used 0.5 Tg SO2 was emitted during this eruption, why not also producing a model run with larger SO2 injections? See also SC12.

The analyses presented in this manuscript are based on the simulations produced within the multi-model intercomparison framework designed by Zhu et al. (2025). At the time these experiments were designed, 0.5 Tg of SO2 was the most widely accepted estimate. The coordinated simulations required substantial computational effort and took approximately one year to complete; therefore, modifications to the emission protocol are not currently feasible. As a result, our analysis is necessarily limited to the scenarios defined within the original intercomparison framework.

**Specific Comments**

SC1) L20-22: Many studies have been realised in the last 10-15 years on the observed series of moderate stratospheric eruptions. This must be discussed here, especially because of the magnitude of the Hunga eruption. For a general context, see e.g. https://www.science.org/doi/10.1126/science.1206027. For specific eruptions with corresponding radiative forcing estimations, see e.g. https://acp.copernicus.org/articles/21/535/2021/ (Raikoke eruption 2019), https://www.nature.com/articles/ncomms8692 (decadal series of recent eruptions) and others.

We have added references to recent studies focusing on moderate volcanic eruptions and rephrased the paragraph to highlight the growing importance of considering these eruptions in climate studies.

"While often studies of volcanic aerosols' impact on climate focus on very large eruptions like Mt. Pinatubo, that erupted in 1991 (Quaglia et al., 2023) releasing anywhere between 10 and 20 Tg-SO2 in a few days (Baran and Foot, 1994; Fisher et al., 2019), recent decades have seen fewer eruptions with injections exceeding a few Tg of SO2 (Carn et al., 2017; Brodowsky et al., 2021). However, moderate eruptions, such as Raikoke (2019), Ulawun (2019), Ambae, La Soufrière (2021), and Hunga Tonga (2022), have received increasing attention in recent research, offering valuable insights into aerosol microphysics, plume dynamics, and radiative forcing at the top of the atmosphere (Kloss et al., 2021; Wrana et al., 2023). Despite their smaller scale compared to Pinatubo, moderate eruptions can still substantially double the quiescent stratospheric sulfate burden (Schmidt et al., 2018; Andersson et al., 2015; Brodowsky et al., 2024), underscoring the importance of including these events in both climate modeling and aerosol research."

SC2) L26: I think the limited interest in ash for this kind of events (moderate eruptions) is its limited atmospheric lifetime. This can be mentioned here.

We agree that ash has a limited atmospheric lifetime in moderate eruptions. However, this point is not directly relevant to the specific sentence, which focuses on the co-emission of volcanic byproducts that may reach the stratosphere. Therefore, we elected not to add this detail here, to avoid distracting from the main point of the paragraph.

SC3) L28-29: I think the right name for the volcano is just "Hunga", which is not an abbreviation. Please correct throughout the text.
Done. However we kept the full name Hunga Tonga–Hunga Ha'apai when referring to the multi-model intercomparison project paper, as it appears in the title of the published study.

SC4) L29: "unprecedented" with respect to what reference? Please be more precise.
We rephrased the sentence as follow: "Hunga erupted on January 15 2022 in the South Pacific, releasing an amount of materials into the atmosphere unprecedented in the satellite era (Carr et al., 2022)"

SC5) L32: The estimation of 1.6 pm 0.5 Tg is for the formed sulphate aerosol mass, not SO2, in Sellitto et al., 2024. In that paper, the authors estimate the SO2 injected mass at values >1.0 Tg (1.0 Tg being the lower limit). Please correct.
Corrected.

SC6) L33: Be careful on the use of the word "minor" here, because you are still talking about volcanic eruptions strong enough to inject into the stratosphere. I would say "recent moderate stratospheric eruptions" or something similar.
We replaced 'minor' with 'moderate'.

SC7) L34: While the water vapour injections for Hunga are unprecedented in terms of satellite observations (in the "satellite era") the debate is still open about the possibility that other eruptions could have injected more water vapour than thought, see
https://www.nature.com/articles/s43247-024-01651-w
We referenced the paper as follows:

"While the amount of injected sulfur was within the range of many moderate past explosive volcanic eruptions, the amount of water vapor co-injected was unprecedented in the historical record, representing roughly ~10% of all background stratospheric water vapor. Although comparable increases in moisture can be achieved through the indirect pathway, such as during Pinatubo-like eruptions in the tropics (Kroll et al., 2024), what makes Hunga eruption truly unprecedented aspect of this eruption was the height at which the overall volcanic plume was injected, reaching up to 55 km (Carr et al., 2022)."

SC8) L35: Not only of the "injected SO2" but the plume overall (SO2, water vapour, short-lived ash and ice – i.e. the overall volcanic plume)
Corrected (see SC7).

SC9) L35-36: The "warming effect" of the water vapour was not just found as an "early estimate" (and, in any case, note that Jenkins et al. don't consider aerosols at all in their estimations) but was due to the specific morphology of the early phases of the Hunga plume's dispersion (rapid descent of the water vapour and sulphate aerosols to altitudes where water vapour is very effective in the longwave, Sellitto et al., 2022). This changed with the following separation of the sulphate aerosols and water vapour plumes (water vapour rose, trapped into the general stratospheric circulation, and sulphate aerosols descended, due to gravitational sedimentation). Please correct this part of text accordingly.

We think it is fair to say that the Jenkins et al. was an "early estimates", in the sense that their estimates was due to, as the reviewer states, "the specific morphology of the early phases of the Hunga plume's dispersion".

Jenkins et al. don't consider the injection of SO2 in their calculations, stating "We ignore the negative IRF contribution from the accompanying SO2 deposit since the SO2 deposit is significantly smaller than the accompanying water vapour deposit, and it is unclear that the SO2's cooling response would be measurable following a HTHH-sized stratospheric SO2 injection." The paper also concludes "the HTHH eruption temporarily does increase the GMST anomaly over the next five years, while stratospheric water vapour concentrations are perturbed. Over this period HTHH increases the likelihood we observe our first 1.5°C year by around 7%."

Based on this, we think that our characterization of the historical evolution of the collective understanding of the HTHH radiative impact is correct.

SC10) L40-42: I think that this part/discussion here is rather incomplete and must be extended, and the main assumption (larger particles = stronger negative radiative forcing) is just wrong. There are actually two main reasons for the larger (negative) radiative efficacity of Hunga's SO2 injections with respect to other recent moderate stratospheric eruptions and larger ones as Pinatubo 1991. First, Li et al. (2024) (https://agupubs.onlinelibrary.wiley.com/doi/10.1029/2024GL108522) shown that Hunga's SO2 emissions produce larger stratospheric aerosol extinction per unit emitted SO2 mass than recent major eruptions, such as the one of Pinatubo in 1991, due to the specific aerosol size distribution in the Hunga plume and the high-altitude SO2 injection. Second, Sellitto et al. (2025) (https://acp.copernicus.org/articles/25/6353/2025/) shown that the specific size distribution of the formed Hunga's sulphate aerosols are more radiatively effective towards negative radiative forcing than Pinatubo etc, considering the shortwave radiative forcing efficiency dependency on the effective radius and the shortwave-to-longwave individual radiative interactions (see Fig. 7 therein). In both cases, the larger efficiency of Hunga's aerosols with respect to e.g. Pinatubo is not because they are larger in size but rather because they are smaller! (see Sellitto's Fig. 7 and Li's Fig. 3b). Please correct and elaborate the text with these elements.

We would like to clarify that our manuscript does not claim that larger particles directly imply a larger radiative forcing. Rather, we state that 'large quantities of water vapor favored particle

growth in the initial months' (Sect. 3.1) meaning growth relative to background conditions, sufficient to reach sizes that are optically efficient at scattering radiation. To avoid any potential ambiguities, we have rephrased the corresponding sentence in the manuscript and now cite Zhu et al. (2022), Li et al. (2024), and Sellitto et al. (2025) together, as their results are consistent and complementary. The revised sentence as follows:

"Notably, the conditions of significant stratospheric hydration has been suggested as an important factor contributing to the substantial radiative impact of the Hunga plume, even in the case of a modest SO2 injection, as it promoted faster aerosol growth, allowing particles to reach sizes that are optically efficient scatterers (Zhu et al., 2022; Li et al., 2024; Sellitto et al., 2025)."

This wording avoids implying that Hunga aerosols are larger than those from other eruptions, while still highlighting that the water-vapor-driven growth leads to radiatively effective particle sizes.

SC11) L65-68: The very peculiar dynamics of the plume during the very first weeks of dispersion after the main eruption (injection at very high altitude and then fast radiatively-driven descent due to longwave cooling due to water vapour, Sellitto et al., 2022) must be mentioned here. How is this considered in your injection modelling?
We added the following sentence explaining how models simulate the initial descent driven by water vapor cooling:

"Models account for the interactive coupling between aerosol, water vapor, radiation, and dynamics, allowing the fast descent due to water vapor longwave cooling (Sellitto et al., 2022) to be simulated through this coupling."

SC12) Table 1: Please explain why simulations with larger SO2 total mass injections (suggested by later satellite observations) were not realised in this work.
The answer is included in the response to comment MC4.

SC13) L83: "form" --> "from", I guess.
Corrected.

SC14) L169-170: I don't agree that "larger particle grow" results in "a larger radiative forcing", see previous comments. Also, can you please show the evolution of the obtained size distributions in your modelling with and without H2O, please? How so they compare with the observations of e.g. Duchamp et al., 2023 (https://agupubs.onlinelibrary.wiley.com/doi/full/10.1029/2023GL105076) and Boichu et al., 2023 (https://agupubs.onlinelibrary.wiley.com/doi/full/10.1029/2023JD039010)?

We answer the first part of the comment in SC10. Regarding the request to show the evolution of modeled size distributions with and without H2O and to compare them with observations, such an analysis lies outside the scope of the present study. This paper focuses specifically on the radiative response to the Hunga eruption among models and within three different meteorological configurations, not on a detailed microphysical evaluation or an observational

intercomparison. These aspects are addressed in the Hunga report and will be discussed more comprehensively in a dedicated forthcoming paper.

SC15) L171-172: "...may suggest...for the two cases": this is all very speculative. Is there any way to test how these processes contribute to the size distributions in the two cases?

We added a new figure (Fig. A5) showing the time series of the tropical effective radius, where available, for the SO2andH2O, SO2only, and NoVolc. This figure highlights the faster initial particle growth and more rapid return to background values in SO2andH2O compared to SO2only. In addition, we have included a figure showing the stratospheric AOD (Fig. A3). We also recognized that the original sentence in the first bullet point was not fully appropriate, and we have revised it and now clarify this aspect when discussing Fig. 3, as follows:

"The IRF from the SO2andH2O and SO2only simulations (black and orange lines in panel 3a) show similar behavior from late 2022 onward. However, substantial differences are evident during the first six months (on the order of 20% in WACCM6-MAM, panel 3c), with the exact duration being model dependent (Fig. A4). This is in agreement with previous studies indicating that the co-injection of stratospheric water vapor promotes faster particle growth to optically efficient scattering sizes during the initial months (panels a–c for SO2andH2O versus panels d-f for SO2only in Fig. A5). As a result, SO2andH2O produces a larger initial stratospheric AOD (Fig. A3 j-k) and corresponding negative radiative forcing. Later on, the forcing becomes similar due to a compensating effect: SO2only shows a larger stratospheric AOD, while SO2andH2O shows a stronger gas contribution (negative at TOA). The SO2andH2O forcing becomes slightly weaker only in the last few months of 2023, when the gas IRF in the two simulations becomes comparable (panel 3c)."

[Figure]

**Change in stratospheric AOD at 550 nm**

Figure A3. Time series of zonal mean stratospheric aerosol optical depth anomalies (perturbed minus control) from five models: WACCM6-MAM (a, f), WACCM6-CARMA (b, g), MIROC-CHASER (c, h), CAM5-CARMA (d, i), and UKESM (e). The first two rows correspond to perturbation scenarios from SO2andH2O, and the second row shows SO2only. The third row includes regional averages of the same quantity for two latitude bands: 90°S-N (j), 60°S–equator (k) and 60–90°S (l).

[Figure]

Figure A5. Time series of effective radius averaged over 30°S and the equator for (a, d, g) WACCM6-MAM, (b, e, h) WACCM6-CARMA, and (c, f, i) MIROC-CHASER. Each row corresponds to a different scenario from the nudged experiment: the first column shows SO2andH2O, the second column SO2only, and the third column NoVolc.

SC16) L175: when you say "stratospheric gases", which gases do you mean, except for H2O? In the Radiative forcing section, we added a sentence that explains what gases are included in the double radiation call.

"In WACCM6-MAM, a second estimate of instantaneous radiative forcing for aerosols (sulfate, black carbon, primary organic matter, secondary organic aerosols, sea salt, and dust) and gases (water vapor, dioxygen, carbon dioxide, ozone, nitrous, methane, chlorofluorocarbons) is derived through a double radiation call [...]"

However, the main stratospheric gases affecting the radiative forcing are stratospheric water vapor and ozone, and now we explicitly discuss their roles in the Results sections.

SC17) L176-178: Again, the H2O impact is very dependent on the altitude of the H2O plume (https://www.science.org/doi/10.1126/science.1182488). This should be here mentioned, in particular with respect to the initial large variability of the plume height in the first month of dispersion, as shown by Sellitto et al., 2022.
We added a sentence at the end of the bullet point, but we found it more appropriate to cite Millan et al. (2022) and Zhuo et al. (2025).

"The small impact of stratospheric water vapor is altitude-dependent, with the largest effect occurring near the tropopause rather than at higher altitudes (Solomon et al., 2010) Indeed, stratospheric water vapor concentrations are primarily confined to the middle stratosphere (40–10 hPa) by the end of 2022, after which they rise and extend into the lower mesosphere (Millan et al., 2022; Zhuo et al., 2025)."

SC18) L189: maybe just a detail but your outputs are, I imagine, discretised: so, you are not calculating the moving average forcing as actually an integral, isn't it?
Yes, therefore we discretized the equation as follows:

"[..] the moving average forcing, calculated as $\frac{1}{N_t}\sum_{i=1}^{N_t} RF_i$, where $N_t$ represents the number of months elapsed, and $RF_i$ is the radiative forcing at month i. "

SC19) L192: How a moving average forcing can be a measure of the "cumulative" forcing? I may be missing something here, but I don't agree with this. Can you please explain?
We understand that the term "cumulative" may have been misleading, so we have revised the sentence as follows:
"We use the moving average forcing instead of the instantaneous forcing to highlight the time-averaged impact of the eruption over time, which provides a better measure of its overall climatic effects."

SC20) L198-202: Please mention here where this can be seen (panels of Fig. 4). Also, is this truly so different (for me, the magnitude and the overall spatiotemporal evolution of sAOD between the two model runs is quite similar). And if you really think it is significantly different, why the ERF response is so consistent? Please explain.

We have referenced the respective panels of Fig. 4 when discussing the changes in stratospheric AOD, water vapor, and ozone. Additionally, we have expanded the explanation after addressing these changes. However, the answer remains speculative, as for MIROC-CHASER, we are unable to evaluate the contribution of each individual forcing agent. To clarify, we have added the following section:

"As with the instantaneous radiative forcing, the gas response varies depending on the atmospheric level at which it is calculated and the vertical distribution of the gases. Additionally, temperature adjustments are included here. We speculate that the stronger stratospheric AOD anomaly in WACCM6-MAM, along with the stronger water vapor and ozone anomalies in MIROC-CHASER, may either offset the radiative forcing at the top of the atmosphere (TOA) or

amplify it at the tropopause (TROP). This could explain the similar forcing observed in Fig. 4a and the larger differences seen in Fig.4b. This aspect is only further explored in Sect. 3.3 for WACCM6-MAM, where the availability of separate aerosol and gas radiative contributions allows for a clearer disentangling of each factor."

SC21) Figure 3: which are the "gas" species in this plot?
This is answered in SC16.

SC22) L215-217: I always had the impression that the SAGE-based GloSSAC time series is quite low, in terms of the sAOD for Hunga, than other spatiotemporally denser observations, like OMPS-LP. I strongly suggest making a similar comparison also with the OMPS-LP data set.
We answered this point in Main Comment #3.

SC23) L219: "...while GloSSAC does not see the high peak at the beginning...", as discussed before, this is clearly due to the scarce spatiotemporal sampling of SAGE III/ISS observations, which is the basis of the GloSSAC. Please add a plot with OMPS-LP, more pertinent to study the initial peak in sAOD.

Based on our response to Major Comment #3, we kept the comparison with GloSSAC, refer the reader to Zhuo et al. (2025) for a more detailed discussion including other observational datasets, and explicitly highlight the limitation of GloSSAC in capturing the initial post-eruption peak. The reasons for our observational choice, as discussed in Major Comment #3, has also been included in a new section "Observational dataset". The section corresponding to L219 has been rephrased as follows:

"In Fig. 4 we also provide a comparison with available observations, with more in depth comparisons presented in Zhuo et al. (2025).  In general, both models show good qualitative agreement with the observations. While GloSSAC does not capture the pronounced peak immediately following the eruption, which is evident in other observational datasets discussed in Zhuo et al. (2025), this mainly reflects the limited spatiotemporal coverage of the SAGE III/ISS observations on which GloSSAC is based during the earliest post-eruption phase. Despite this limitation, transport toward the Southern Hemisphere occurs on similar timescales in both the observations and the simulations, with WACCM6-MAM showing a better match in terms of the residual aerosol plume around 60°S."

SC24) Section 3.3: In my opinion, the scopes of the results shown in this section should be better explained, so that this is accessible, in terms of motivations, for the broader readership of ACP. The fact that many multi-panel figures (4) follow with little text explaining their content and their scopes in the paper narrative does not help. The text in this section should be developed further, I think.

We agree with the reviewer and have substantially revised this section to improve clarity and logical flow. In addition, we reordered the discussion of the figures and relocated some figures between Section 3.3 and the Supplementary Material. The revised section is attached below.

"In this section we show a comparison of radiative forcing in the three different experiments, which only WACCM6-MAM conducted in full. This comparison provides useful insights to the different ways to define the radiative impacts of the plume between RF, ERF and IRF.

Figure 5 shows that, across all experiments and at the three atmospheric levels, the simulated clear-sky radiative forcing is negative and locally statistically significant in the Southern Hemisphere during the first two years following the eruption. The sign, as well as the spatial and temporal evolution, is consistent across the different model configurations and atmospheric levels considered. However, when including the atmospheric temperature adjustments in Fixed-SST and the ocean response in Coupled, the radiative forcing from aerosols and water vapor is substantially smaller than in the nudged simulation, particularly at TOA. This difference is more evident when averaging over the Southern Hemisphere (60◦S to the equator), as shown in Fig. 6.

In panel 6a, the clear-sky IRF at TOA during the first 2 years after the eruption is -0.43 W m−2 in the nudged simulations, whereas in the free running experiments it is -0.27 W m−2 in both the coupled ocean and atmosphere-only cases. Although smaller than in the nudged simulations, these responses remain outside the range of natural variability, which is estimated in Fig. 6 as one standard deviation from the control ensemble (NoVolc). In general, while most of the results presented here are for the SH only, when considering global mean, the values are reduced by approximately half, reaching a value of -0.09 W m−2 at the TOA for the Coupled experiment in 2022-2023, which falls within the range of natural variability (see Fig. A7 and Table 4).

The reduced net radiative forcing relative to the nudged case arises from the inclusion of temperature and ozone adjustments. Indeed, differences in the simulated stratospheric AOD changes among the experiments are negligible (Fig. A8a–c), consistent with previous studies showing that atmospheric nudging does not necessarily improve the representation of the residual stratospheric circulation (Chrysanthou et al., 2019). Instead, the response is largely dominated by gas-driven variability, which is strongest in the Coupled experiment and includes a late-period negative radiative forcing in 2025 that is not present in the other simulations (Fig. 6a-b).

The separation of aerosol–radiation and gas–radiation contributions to the total radiative forcing (Fig. 7) shows that the negative TOA forcing during the first two years following the eruption is primarily driven by the aerosol contribution from Hunga, partially offset by a positive contribution from changes in gases. This behavior contrasts with the nudged simulations,  where the gas–radiation interaction produces a negative forcing (Fig. 3). In the free-running simulations, the gas contribution is strongly influenced by natural variability (red line in Fig. 7), particularly for all-sky radiative forcing, which limits the detectability of the forced response without large ensemble sizes (Fig. A9).

In the second half of 2025, as the tropical stratospheric AOD returns to background levels (Fig. A8a–c), only the Coupled experiment exhibits a significant negative radiative forcing in the tropics at both TOA and TROP, with magnitudes comparable to those observed during the first post-eruption year (Fig. 6a and b), while the forcing at SURF becomes positive. This late-period

forcing in the Coupled experiment is primarily driven by gas–radiation interactions, with significant negative clear-sky values emerging in 2025 (Fig. 7c).

The gas radiative forcing results not only from Hunga-induced changes in stratospheric water vapor (which are similar across the three experiments; Fig. A8d–f) but also from changes in stratospheric temperature and ozone, as well as associated dynamical adjustments that modulate both, particularly in the lower stratosphere. These contributions are further explored in Fig. 8. The gas–radiation interaction is dominated by the LW component (Fig. 8a–c), with differences among the atmospheric configurations driven primarily by temperature and dynamical responses: Zhuo et al. (2025) has shown an upper-level cooling due to the water vapor and a lower stratospheric level warming from LW absorption from the sulfate aerosols in the free-running simulations.

During the first two years following the eruption, the TOA gas radiative forcing is positive in the Coupled simulation, more variable in Fixed-SST, and negative in the Nudged case (Fig. 8a–c). This behavior reflects the combined effects of increased water vapor in the middle stratosphere and reduced ozone in the lower stratosphere. When temperature and dynamical changes are excluded, as in the Nudged simulation, the water vapor increase and ozone decrease (black curves in Fig. 8f and i) enhance  outgoing LW radiation at TOA, resulting in a negative forcing. In contrast, when temperature adjustments are included, as in the Coupled and Fixed-SST simulations, the cooling at higher altitudes where water vapor anomalies peak (black curves for water vapor and red curves for temperature in Fig. 8d and e) reduces outgoing LW radiation, yielding a positive TOA forcing during the first two post-eruption years. Furthermore, the Coupled shows a distinct evolution in the tropical lower stratosphere, characterized by increased ozone and warming during 2022–2023, followed by decreased ozone and cooling in 2025 (panel 8g). Because ozone in the lower stratosphere acts as a greenhouse gas, these changes contribute to an additional positive radiative forcing during the first two years and a negative forcing in 2025, when water vapor anomalies have largely returned to background levels. The close correlation of changes in ozone and temperatures in this region, not seen in Fixed-SST (panel 8h), is strongly indicative of their dynamical origin, suggesting an associated decrease in tropical upwelling in 2022-2023 and increase in 2025.

As discussed in (Bednarz et al., 2025), the coupled ocean WACCM6-MAM simulations shows a significant modulation of the El Niño–Southern Oscillation (ENSO) variability by the eruption, with La-Nina like response in 2022-2023 and an El-Nino like response in 2025. In general, ENSO is an important driver of interannual variability in tropical upwelling, which in turn modulates lower-stratospheric temperatures and ozone (Randel et al., 2009), and can therefore exert a substantial influence on the overall radiative forcing. "

SC25) L277: "a small amount of SO2": we still don't know exactly how much SO2 was injected and, strictly speaking, even 0.5 Tg of SO2 is not a "small amount". And "small" with respect to what? Please contextualise this.
Based on https://doi.org/10.1029/2018JD028776, we changed "small" to "small-to-moderate amount".

SC26) L279: this was also obtained with different information, e.g. satellite and RTM, in Sellitto et al., 2025, which should be mentioned here (so that there is also a reference in the "observational world").
Added.

SC27) L280-281: "aerosol size" larger than which explosive eruption? Still smaller than Pinatubo, and the stronger negative forcing efficiency of Hunga than Pinatubo depends on this, see Major Comment 1. Please rephrase.
A more complete answer was given in MC #2. We understand that the sentence was misleading and rephrase it as follows:

"Our multi-model results confirm previous analyses from Zhu et al. (2022), Stenchikov et al. (2025) and (Sellitto et al., 2025) which indicate a potential net negative forcing from the volcanic cloud, due to the formation of a persistent layer of stratospheric sulfate aerosol that rapidly grew to optically efficient sizes, enhancing shortwave scattering relative to background conditions."

SC28) L284-287: Why not starting this paragraph with an introduction of the sAOD in your runs?
The discussion of sAOD in this manuscript is intentionally limited, as it is only included to support our main findings. For this reason, we consider a detailed introduction of sAOD at the beginning of this paragraph unnecessary.

SC29) L300-301: "This suggests that the results...might be underestimated...". Still, this sounds not totally correct because the model results suggest similar sAOD burdens than observations - to be confirmed with the comparison with OMPS-LP data.
This is shown in Zhuo et al. 2025. We added a sentence to refer to that paper:

"This suggests that the results presented here might be underestimated, if higher retrieval estimates for the sulfate aerosol burden were confirmed, particularly in light of the uncertainties in stratospheric AOD retrievals highlighted by comparisons between the GloSSAC and OMPS-LP datasets in Zhuo et al. (2025)."

SC30) L303-304: "However, this relationship..." this is shown by Sellitto et al., 2025
Added.

SC31) L311-313: These points mentioned by Li et al. (2024) must be discussed much earlier in the text and are just half of the story, see Major Comment 1.
We have revised the structure of the Conclusions to improve clarity and logical flow. However, we consider the reference to Li et al. (2024) to be most appropriate in this context.